# Cryptic developmental events determine medulloblastoma radiosensitivity and cellular heterogeneity without altering transcriptomic profile

Daniel Shiloh Malawsky [1,9], Seth J. Weir[1,9], Jennifer Karin Ocasio[1,2], Benjamin Babcock [1], Taylor Dismuke[1], Abigail H. Cleveland [1,3], Andrew M. Donson[4,5], Rajeev Vibhakar[4,5], Kirk Wilhelmsen [1,6,7] & Timothy R. Gershon [1,2,8]

It is unclear why medulloblastoma patients receiving similar treatments experience different outcomes. Transcriptomic profiling identified subgroups with different prognoses, but in each subgroup, individuals remain at risk of incurable recurrence. To investigate why similar-appearing tumors produce variable outcomes, we analyzed medulloblastomas triggered in transgenic mice by a common driver mutation expressed at different points in brain development. We genetically engineered mice to express oncogenic *SmoM2*, starting in multipotent glio-neuronal stem cells, or committed neural progenitors. Both groups developed medulloblastomas with similar transcriptomic profiles. We compared medulloblastoma progression, radiosensitivity, and cellular heterogeneity, determined by single-cell transcriptomic analysis (scRNA-seq). Stem cell-triggered medulloblastomas progressed faster, contained more OLIG2-expressing stem-like cells, and consistently showed radioresistance. In contrast, progenitor-triggered MBs progressed slower, down-regulated stem-like cells and were curable with radiation. Progenitor-triggered medulloblastomas also contained more diverse stromal populations, with more *Ccr2+* macrophages and fewer *Igf1+* microglia, indicating that developmental events affected the subsequent tumor microenvironment. Reduced mTORC1 activity in *M-Smo* tumors suggests that differential *Igf1* contributed to differences in phenotype. Developmental events in tumorigenesis that were obscure in transcriptomic profiles thus remained cryptic determinants of tumor composition and outcome. Precise understanding of medulloblastoma pathogenesis and prognosis requires supplementing transcriptomic/methylomic studies with analyses that resolve cellular heterogeneity.

[1] Department of Neurology, University of North Carolina School of Medicine, Chapel Hill, NC, USA. [2] UNC Neuroscience Center, University of North Carolina School of Medicine, Chapel Hill, NC, USA. [3] UNC Cancer Cell Biology Training Program, University of North Carolina, Chapel Hill, NC, USA. [4] Department of Pediatrics, University of Colorado Anschutz Medical Campus, Aurora, CO, USA. [5] Morgan Adams Foundation Pediatric Brain Tumor Research Program, Children's, Hospital Colorado, Aurora, CO, USA. [6] Department of Genetics, University of North Carolina School of Medicine, Chapel Hill, NC, USA. [7] RENCI, Chapel Hill, NC, USA. [8] Lineberger Comprehensive Cancer Center, University of North Carolina School of Medicine, Chapel Hill, NC, USA. [9] These authors contributed equally: Daniel Shiloh Malawsky, Seth J. Weir. ✉email: kirk@med.unc.edu; gershont@neurology.unc.edu

Medulloblastoma, the most common malignant pediatric brain tumor, is typically treated with surgery, radiation, and chemotherapy, which are effective in 80–90% of patients. However, individual patients face a significant risk of treatment failure despite uniform treatment, and the causes of treatment failure are incompletely understood. Transcriptomic studies have identified four major subgroups of medulloblastoma: WNT, SHH, group 3, and group 4[1]. Each subgroup has a different prognosis, but within each subgroup, outcomes for individuals are heterogeneous[2]. The factors that determine the variable outcomes for patients with similar-appearing medulloblastomas are unclear. We tested the possibility that different outcomes of medulloblastoma therapy can be determined by developmental events that are cryptic at the time of presentation.

SHH-subgroup medulloblastomas are grouped together because they show similar patterns of gene expression in bulk transcriptomic studies, indicating SHH pathway activation. Despite sharing a common oncogenic pathway, patients with SHH-subgroup medulloblastomas show different responses to treatment, with ~20% developing incurable recurrence. It is not clear whether differences in outcome are stochastic or driven by determinants that remain to be identified. Age of onset, however, is a factor that clearly influences prognosis within each subgroup[2], suggesting a potential effect of developmental timing on tumor behavior.

Prior studies in genetically engineered mice show that cerebellar granule cell progenitors (CGNPs) are proximal cells of origin for SHH-driven medulloblastoma[3,4]. CGNPs are a population of SHH-responsive, committed neural progenitors that derive from the rhombic-lip, migrate to the external granule layer (EGL) of the cerebellum, and then proliferate rapidly in response to SHH ligand secreted by the Purkinje neurons[5,6]. CGNPs proliferate in the first two weeks of life in mice, or the first year of life in humans, to generate the cerebellar granule neurons (CGNs) the largest neuronal population in the brain[7]. Mutations that hyperactivate SHH signaling in CGNPs cause familial medulloblastoma in humans and recapitulate medulloblastoma formation in mice, providing genetically faithful primary tumor models[3,4,8].

Importantly, CGNPs are not a homogeneous population. While the EGL is predominantly populated by ATOH1 (AKA MATH1)-expressing progenitors, a small subset of NESTIN+/ATOH1- cells reside in the EGL[9]. These EGL cells are typically quiescent in vivo but proliferate in response to SHH pathway activation and can give rise to SHH-driven medulloblastoma[9]. Moreover, the ATOH1+ cells of the EGL comprise different subsets[10,11], including a transient subpopulation that expresses the stem cell marker SOX2 and may be particularly vulnerable to SHH-driven tumorigenesis[11]. CGNPs are thus a heterogeneous population with varying oncogenic potential.

The developmental origins of medulloblastoma can be analyzed by pairing different Cre drivers with conditional mutations of the SHH receptor components *Ptc* and *Smo* directing SHH hyperactivation to broad lineages that include CGNPs or more narrow lineages within the CGNP population. For example, *hGFAP-Cre* targets a lineage of neuroglial stem cells throughout the brain that includes both ATOH1+ CGNPs and NESTIN+/ATOH1− cells of the EGL[12,13]. *Math1-Cre*, in contrast, targets the ATOH1-expressing CGNPs[14–16], including the SOX2+ subset[11]. A prior study compared the effects of deleting *Ptc* either in neural stem cells in *hGFAP-Cre/Ptc^{loxP/loxP}* mice or in CGNPs in *Math1-Cre/Ptc^{loxP/loxP}* mice[17]. Both genotypes developed medulloblastoma with 100% penetrance and in *hGFAP-Cre/Ptc^{loxP/loxP}* where SHH was hyperactivated throughout the brain, tumors developed only in the cerebellum. Similarly, inducing a Cre-dependent, constitutively active allele of *Smo* (*SmoM2*) in either stem cells or CGNPs, using respectively *hGFAP-Cre* or

*Math1-Cre*, resulted in medulloblastoma with 100% penetrance and no other brain tumors[18]. These studies show that the EGL population is uniquely competent to undergo SHH-mediated tumorigenesis and that cells with hyperactivation of SHH signaling prior to the formation of the EGL must advance along the CGNP developmental trajectory by migrating to the cerebellar surface before giving rise to tumors[17,18].

Medulloblastomas initiated by *hGFAP-Cre* in stem cells progress faster than medulloblastomas initiated by *Math1-Cre*, producing a shorter EFS despite occurring in the same location and showing similar pathology and gene expression profile[17,18]. Similarly, medulloblastomas initiated by conditional activation of *SmoM2* either prenatally, using *hGFAP-Cre*, or postnatally, using tamoxifen-inducible *Math1-CreER*, are histologically and molecularly indistinguishable, but show different propensities for anchorage-independent growth in vitro[19]. In these studies, SHH-driven medulloblastomas triggered early, in stem cells or later, in progenitor cells, show overall similarities but also specific differences.

The different survival times when tumors are triggered with *hGFAP-Cre* or *Math1-Cre* suggest that the timing of the oncogenic event can act as a cryptic factor that produces clinically relevant effects that persist throughout the generations of tumor cells. We tested the clinical relevance of this possibility by comparing the responses to therapy of medulloblastomas initiated in either stem cells or neural progenitors, and by subjecting both types of tumors to scRNA-seq analysis. We show that timing of tumor initiation within the lineage trajectory of GFAP+ stem cells to ATOH1+ progenitors influences the cellular heterogeneity within the resulting tumors, without detectably altering average gene expression profiles, producing tumors that appear similar but contain divergent sub-populations with different tumor-stromal interactions and treatment responses.

## Results

### Similar pathology and gene expression in medulloblastomas from progenitors or stem cells.
To initiate an oncogenic stimulus in CGNPs, we bred *SmoM2* mice with *Math1-Cre* mice to generate *Math1-Cre/SmoM2* (*M-Smo*) pups. To initiate an oncogenic stimulus earlier in brain development in pluripotent stem cells that give rise to CGNPs, we bred *SmoM2* mice to *hGFAP-Cre* mice to generate *hGFAP-Cre/SmoM2* (*G-Smo*) pups. Both *M-Smo* and *G-Smo* genotypes developed cerebellar tumors with 100% frequency and all tumors showed the small, round blue cell morphology typical of medulloblastoma and tendency to invade adjacent brain (Fig. 1a).

Microarray comparison of gene expression in samples from 6 *M-Smo* and 6 *G-Smo* tumors showed similar transcriptomic profiles with 64/41349 probes sets detecting statistically significant signals, representing 33 identified transcripts (Supplementary Data 1). One of these differentially expressed transcripts, *Protamine 1* (*Prm1*) was included within the *hGFAP-Cre* transgene, and thus expected to be differentially detected. Finding differential *Prm1* provided an internal validation of the assay while finding only 32 other differential transcripts demonstrated the high similarity between the tumors. Similarly, the same workflow applied to the previously published microarray datasets from *G-Smo* and *M-Smo* tumors[18] identified 66 out of 45105 probe sets, representing 54 differentially detected transcripts (Supplementary Data 2). Only 1 gene was differentially expressed in both studies. Both microarray comparisons demonstrated highly similar average gene expression profiles of *G-Smo* and *M-Smo* tumors, with minimal consistently observed differences.

### Different, clinically relevant behaviors of medulloblastomas from progenitors or stem cells.
In contrast to the similarity in

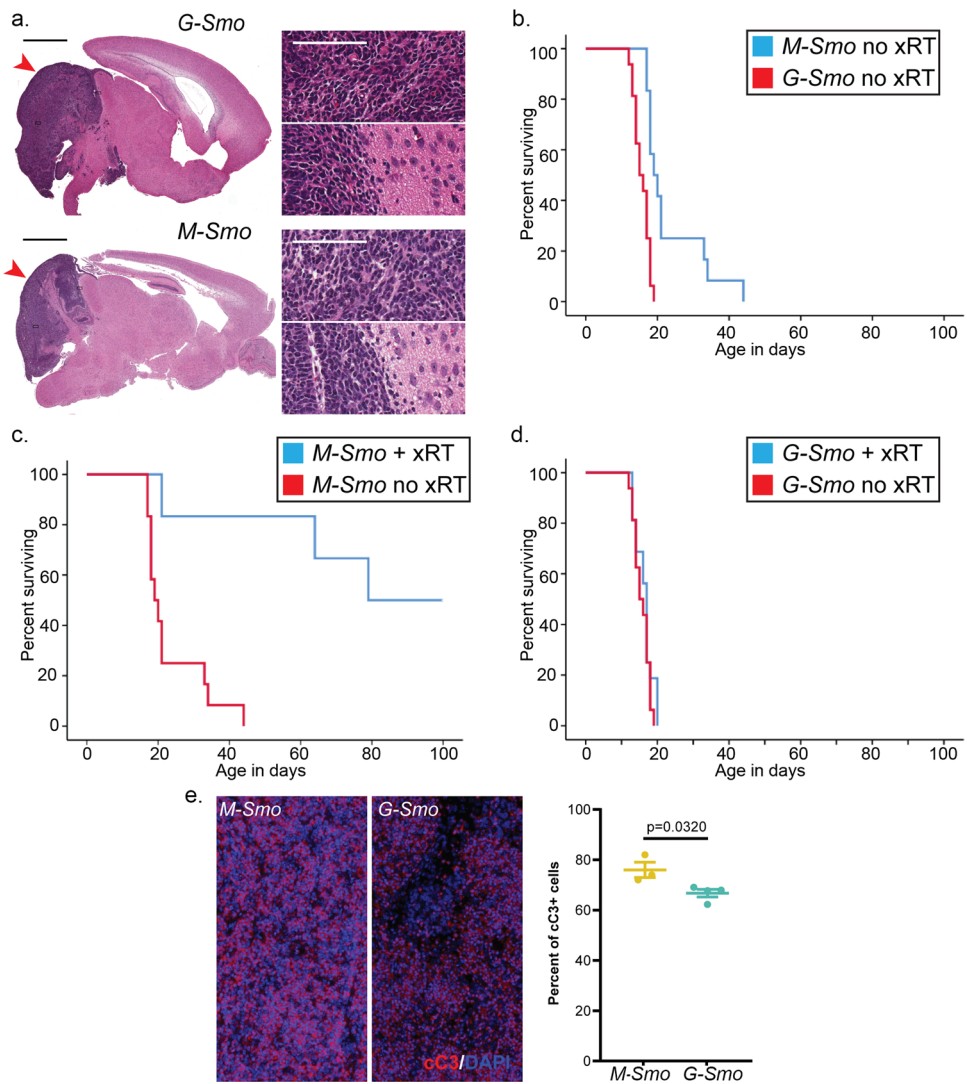

**Fig. 1 M-Smo and G-Smo tumors show similar histology but different rates of progression and responses to treatment. a** Representative sagittal H&E-stained sections of each genotype at P15, with tumors highlighted by red arrowheads. Insets show regions within the tumors and regions at the tumor interface with the adjacent brain. Scale bars = 2 mm, or 100 μm in insets. **b–d** Kaplan–Meier curves comparing **b** survival of untreated mice of each genotype, **c** survival of *M-Smo* mice with and without xRT, and **d** survival of *G-Smo* mice with and without xRT. **e** Representative images of cC3 IHCin the indicated genotypes, 3 hours after xRT, and quantification of replicates. Scale bars = 25 μm. Log-rank test was used to compare survival times in **b–d** and Student's *t*-test was used in **e**. Dots represent values for individual replicates, bars indicate the means, and whiskers indicate the SEM.

pathology and average gene expression of *G-Smo* and *M-Smo* tumors, we noted marked differences in the clinically relevant parameters of survival time and response to therapy. As in the prior study[18], the survival times of untreated *G-Smo* mice were typically shorter than those of *M-Smo* mice (Fig. 1b). Moreover, while *M-Smo* mice showed significantly improved survival after radiation therapy (xRT), consistent with our prior studies[20], xRT did not extend the survival of *G-Smo* mice (Fig. 1c, d). In all these studies, 8–12 mice of each genotype were studied. The prognosis of *G-Smo* and *M-Smo* mice was therefore markedly different, despite their common oncogenic driver, tumor pathology, and similarity in bulk transcriptomic studies.

**Apoptotic response after xRT is less uniform in stem cell-derived medulloblastoma.** To compare the cellular responses to xRT in *G-Smo* and *M-Smo* tumors, we analyzed apoptosis after xRT treatment. We have previously shown that a single fraction of xRT induces widespread, synchronous apoptosis after a 3-h latent period[21], and that this apoptotic response to xRT is

required for efficacy[20]. We treated P15 *G-Smo* ($n = 4$) and *M-Smo* ($n = 3$) mice with 10 Gy xRT delivered in a single dose and then harvested tumors after 3 hours ($n = 3$). We identified apoptotic cells using immunohistochemistry (IHC) for cleaved caspase-3 (cC3; Fig. 1e) and compared the fraction of apoptotic cells in each genotype. We found that xRT induced apoptosis throughout the tumors in both genotypes. However, radiated *G-Smo* tumors showed significantly smaller fractions of apoptotic cells compared to radiated *M-Smo* tumors. While both tumor types showed large radiation-sensitive populations, cells that survived radiation were consistently more numerous in *G-Smo* tumors.

**Similar average gene expression in G-Smo and M-Smo tumors after xRT.** To determine if the differences in cellular responses to xRT produced detectable differences in the transcriptomic profiles in radiated *G-Smo* and *M-Smo* tumors, we analyzed 6 tumors of each genotype harvested 2 h after a single 10 Gy dose of xRT using expression microarrays. We used a single fraction of xRT in

order to produce synchronized changes in gene expression, and harvested tumors after 2 h, in order to study the latent period prior to the onset of apoptosis. 74 probes sets detected statistically significant signals in radiated tumors versus untreated tumors, representing 73 identified transcripts altered by xRT (Supplementary Data 3). Consistent with prior studies[20], xRT induced p53-regulated genes, including *Cdkn1a*, *Trp53inp1*, and *Bbc* (aka *PUMA*). We used 2-way ANOVA to analyze the interaction of genotype and radiation treatment in the combined transcriptomic data from untreated and radiated *G-Smo* and *M-Smo* tumors; we found no genes with differential expression with corrected $p$ value of <0.05. Bulk transcriptomic analysis of treated and untreated *G-Smo* and *M-Smo* tumors thus identified genes induced by xRT, but did not identify genotype-specific differences in transcriptomic response.

**scRNA-seq identifies the difference between medulloblastomas from progenitors or stem cells.** To compare differences between *G-Smo* and *M-Smo* tumors with cellular resolution, we analyzed both tumor types using scRNA-seq. Transcriptomic analysis at single-cell resolution allowed us to examine tumor subpopulations that might be overlooked in bulk transcriptomic studies. We harvested medulloblastomas from 5 *M-Smo* and 6 *G-Smo* at P15, dissociated the tumors, and subjected the cells to bead-based Drop-seq analysis, as previously described[10]. Putative cells identified by bead-specific barcodes were subjected to exclusion criteria described in the "Methods" section, to address the common problems of gene drop out, unintentional cell–cell multiplexing, and premature cell lysis[22,23]. 5930 out of 11984 putative *M-Smo* cells, and 8699 out of 16,489 *G-Smo* cells met the criteria and were included in the analysis. To compare the two genotypes at similar sequencing depths, we randomly downsampled the *G-Smo* transcript counts to 60% of the original depth, as recommended in prior studies[24].

We subjected the scRNA-seq data from *M-Smo* and *G-Smo* tumors to a single principal component analysis (PCA) followed by Louvain clustering, as in our prior studies comparing *M-Smo* tumors with and without treatment[10]. We identified 17 principal components that described >78% of the variance and used UMAP to place cells in a 2-dimensional space according to their distances in PCA space, with Louvain clusters color-coded (Fig. 2a). We noted that cells in the same clusters localized close together in the UMAP, supporting the validity of the cluster assignments. To determine the biological relevance of the clusters, we generated cluster-specific differential gene expression profiles (Supplementary Data 4); we compared for each gene the expression by cells within each cluster to the expression by all cells outside the cluster using Wilcoxon rank-sum test. We then used cluster-specific gene expression patterns to infer the type of cells represented by each cluster.

Using these methods, we identified 8 clusters as different types of stromal cells typical of brain tissue, including astrocytes, oligodendrocytes, macrophages/microglia, vascular cells, fibroblasts, and ciliated cells resembling ependymal or choroid plexus cells (Table 1 and Fig. 2b). These 8 clusters localized as discrete single-cluster units on the UMAP projection. The other 14 clusters localized to a multi-cluster complex in which each cluster shared a border with other clusters. We identified these 14 clusters as tumor cells in a range of states that paralleled CGNP development, from proliferative cells at different phases of the cell cycle to non-proliferative cells at different stages of neural differentiation (Table 1). We identified proliferative clusters by expression of proliferation marker *Mki67*, Cyclin expression, and SHH transcription factor *Gli1*, and further characterized proliferative cells as quiescent, cycling or M-phase enriched

based on cluster-specific gene expression (Table 1). The non-proliferative clusters showed successive expression of early to later differentiation markers *Barhl1*, *Cntn2*, *Rbfox3*, and *Grin2b* (Table 1 and Fig. 2c), as in our prior study of *M-Smo* tumors[10]. We included CGNs as the most differentiated cell type within this group.

To compare the populations within *M-Smo* and *G-Smo* tumors, we deconvoluted the UMAP by genotype (Fig. 2d). For quantitative comparison, we determined the number of cells from each replicate animal in each cluster, normalized to the total number of cells from that animal, and then compared the cluster populations from replicate *M-Smo* and *G-Smo* mice (Fig. 2d, e). We found that most tumor cell clusters were similarly populated in *M-Smo* and *G-Smo* tumors. However, Clusters 1, 2, and 7, within the proliferative region, were significantly enriched in *G-Smo* tumors ($p = 0.008$ for each by Wilcoxon rank-sum test), while cluster 13, comprising CGNs at the differentiated pole, was significantly enriched in the *M-Smo* tumors ($p = 0.023$). Statistically significant enrichment of smaller magnitude was also seen in fibroblast ($p = 0.023$) and differentiated oligodendrocyte clusters ($p = 0.008$) in *M-Smo* tumors.

The similarity in the populations of most clusters in *G-Smo* and *M-Smo* tumors was consistent with the similarity of these tumors in bulk transcriptomic studies. The differential representation of specific types of tumor and stromal cells in *G-Smo* and *M-Smo* tumors, however, demonstrated differences in tumor subpopulations that could not be detected using bulk transcriptomic analysis. *G-Smo* tumors showed increased proliferation, while *M-Smo* tumors showed increased differentiation, and were specifically depleted in the cell types represented by clusters 1, 2, and 7. The expression patterns of all genes detected in our studies can be plotted and compared in *G-Smo* and *M-Smo* UMAPs through our web-based application: https://gsmovmsmoviewer.shinyapps.io/GvMviewer/.

**G-Smo tumors show larger populations of cells expressing stem cell markers.** To characterize further clusters 1, 2, and 7 that were composed predominantly of *G-Smo* cells, we defined the set of genes differentially upregulated in these clusters compared to all tumor cells in *M-Smo* mice, excluding stromal cell types (Supplementary Data 5). We excluded stromal cells in order to prevent stromal gene expression patterns from obscuring differences in tumor cell gene expression. We noted that Clusters 1, 2, and 7 showed increased expression of genes associated with stem cell phenotype, including *Nes*, *Vim*, *Olig1*, and *Olig2*. Feature plots of each of these genes confirmed increased expression in *G-Smo* tumors, particularly in the region of Clusters 1, 2, and 7 (Fig. 2f). We selected *Olig2* for further study because recent functional genetic studies have shown that *Olig2*+ tumor cells in medulloblastoma are cancer stem cells that play an important role in medulloblastoma initiation and recurrence[25].

**Different temporal patterns of stem cell behavior in M-Smo and G-Smo tumors.** To confirm the differential expression of *Olig2* at the protein level and to compare the temporal course of *Olig2* expression efficiently between genotypes, we labeled tumor sections using IHC. Our scRNA-seq data showed that both oligodendrocytes and tumor stem cells express *Olig2*, and that oligodendrocytes can be distinguished from stem cells by the expression of *Sox10* (Fig. 2b, f). Labeling tumor sections with OLIG2 and SOX10 antibodies (Fig. 3a) demonstrated both OLIG2+/SOX10+ cells that we considered to be oligodendrocytes and OLIG2+/SOX10− cells that we considered to be OLIG2-expressing tumor stem cells, equivalent to the *Olig2*+ cells of clusters 1, 2, and 7. The OLIG2/SOX10 assay allowed us to

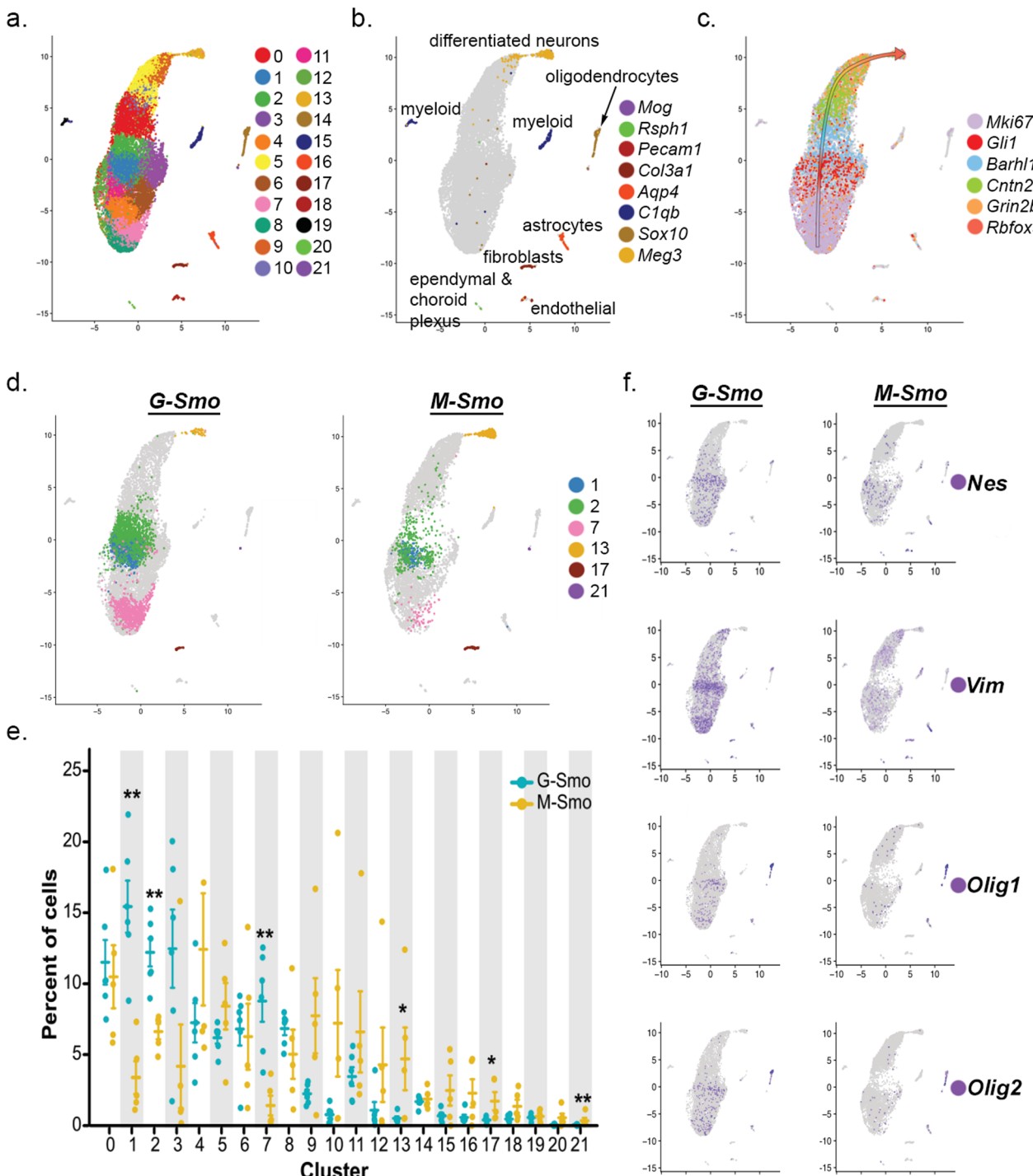

**Fig. 2 Similarities and differences in the sub-populations of G-Smo and M-Smo tumors. a** UMAP plot of all cells from *G-Smo* and *M-Smo* tumors, color-coded by cluster. Cells are localized according to their proximity in PCA space. **b, c** Feature plots showing expression of **b** stromal markers, or **c** cerebellar granule neuron differentiation markers, color-coded over the UMAP shown in **a** arrow in **c** indicates the direction of differentiation across the UMAP. **d** UMAP plots deconvoluted by genotype, with differentially represented clusters color-coded. **e** Comparison of each cluster population in *G-Smo* and *M-Smo* tumors. Dots represent values for individual replicates, bars indicate the means and whiskers indicate the SEM. ** indicates *p* < 0.01, * indicates *p* < 0.05; Student's *t*-test was used to make the pairwise comparison. **f** UMAP plots for each genotype, showing expression of the indicated stem cell markers.

differentiate tumor stem cells from oligodendrocytes and to compare tumor stem cell populations at different ages and treatment conditions (Fig. 3b)

Quantitative analysis confirmed increased OLIG2+ stem cells in P15 *G-Smo* tumors compared to *M-Smo* tumors (Fig. 3c). Differences in OLIG2+ stem cell populations could be attributed to differences in stem cell proliferation, or to differences in the

tendency to maintain a stem cell phenotype. We compared proliferation in the stem cell populations of P15 *G-Smo* and *M-Smo* tumors to determine whether the production of stem cells differed between genotypes. We measured stem cell proliferation using IHC for OLIG2 and the proliferation marker phosphorylated RB (pRB; Supplementary Fig. 1a). Quantitative analysis showed no significant difference in proliferation rate in the

**Table 1 Identification of clusters as specific types of tumor and stromal cells.**

| Cluster | Cell-type designation | Distinctive markers |
|---|---|---|
| 0 | Early differentiating CGNP-like tumor cells | *Pde1c, Nhlh1/2* |
| 1 | Proliferative, quiescent tumor cells | *Hes1, Ccnd1* |
| 2 | Proliferative, quiescent tumor cells | *Srebf1, Ccnd2* |
| 3 | Proliferative, quiescent tumor cells | *Srebf1, Ccnd1* |
| 4 | Proliferative, cycling tumor cells | *Top2a, Lig1, Esco2* |
| 5 | Differentiating CGNP-like tumor cells | *Mtss1, Cntn2* |
| 6 | Proliferative, cycling tumor cells | *Hells, Rrm2* |
| 7 | Proliferative, cycling tumor cells, M-phase enriched | *Cdc20, Cenp1* |
| 8 | Proliferative, cycling tumor cells, M-phase enriched | *Aspm, Ccnb1* |
| 9 | Late differentiating CGN-like | *Pcp4, Car10* |
| 10 | Differentiating CGN-like tumors | *Cntn2, Nhlh1* |
| 11 | Proliferative, cycling tumor cells | *Hells, Lig1, Pclaf* |
| 12 | Proliferative, cycling tumor cells | *Hells, Lig1, Gli1* |
| 13 | CGNs and CGN-differentiated tumor cells | *Gabra6, Vsnl1* |
| 14 | Oligodendrocytes | *Ptpz1, Sox10, Fabp7, Olig1/2* |
| 15 | M2 microglia/macrophages | *Mrc1, C1qb, Aif* |
| 16 | Astrocytes | *Aldoc, Aqp4, Fabp7* |
| 17 | Fibroblasts | *Lum, Vtn* |
| 18 | Endothelial cells | *Cldn5, Flt1* |
| 19 | M1 microglia/macrophages | *Selplg, Siglech, C1qa, Aif* |
| 20 | Ependymal/choroid plexus cells | *Rsph1, Folr1* |
| 21 | Myelinating oligodendrocytes | *Opalin, Plp* |

OLIG2+ populations (Supplementary Fig. 1b). These data indicate that differences in stem cell populations in P15 *G-Smo* and *M-Smo* tumors reflect different tendencies to maintain OLIG2+ stem cell phenotype, rather than differences in stem cell production.

Prior studies found that OLIG2+ tumor stem cells were more numerous in the early stages of medulloblastoma tumorigenesis and decreased as tumors enlarged[18]. We, therefore, compared stem cell dynamics in *G-Smo* and *M-Smo* tumors at P5, early in tumor formation, and at P15 when tumors were larger, using 3–5 replicates of each genotype at each age. We found that at P5, unlike P15, both *M-Smo* and *G-Smo* comprised similarly large fractions of OLIG2+/SOX10− cells (Fig. 3b, c). The OLIG2+/SOX10− fraction decreased over time in both genotypes, but the decrease was more marked in *M-Smo* tumors (Fig. 3b, c).

We explored whether the stem cell dynamics in response to external stimuli were different in *G-Smo* and *M-Smo* tumors. Prior studies showed that sub-curative cytotoxic treatment of medulloblastoma with radiation or chemotherapy induces proliferation of stem cells in the perivascular niche, and that these cells express OLIG2[25,26]. We compared the tumor stem cell populations in *G-Smo* and *M-Smo* tumors recurring after treatment. We subjected P11 *G-Smo* and *M-Smo* mice to cytotoxic treatment with etoposide at a dose calibrated to produce regression followed by recurrence, and then quantified OLIG2+/SOX10− cells 4 days later, at P15. Compared to untreated tumors at P15, we noted increased expression of OLIG2 in SOX10− cells, particularly in perivascular regions, and quantitative analysis showed that etoposide induced a significant increase in stem cells in *M-Smo* tumors, to levels similar to untreated P15 *G-Smo* tumors, while stem cell populations in G-Smo tumors did not change significantly (Fig. 3b, c). The stem cell populations in *M-Smo* tumors showed a greater tendency for

dynamic change and varied inversely with tumor size, declining over time as tumors grew, and increasing after tumor shrinkage imposed by etoposide. In contrast, stem cell populations in *G-Smo* tumors were less dynamic, showed a smaller decline over time, and varied less across all conditions tested. The difference in the developmental timing of oncogenesis in *G-Smo* and *M-Smo* tumors thus continued to affect stem cell regulation weeks after the oncogenic events.

OLIG2-expressing stem cells have been shown to be treatment-resistant in mouse medulloblastoma models, and to drive recurrence[25]. Our findings that non-apoptotic cells were more numerous in *G-Smo* tumors after xRT and that *G-Smo* tumors harbored larger OLIG2+ stem cell populations suggest that the increased OLIG2+ stem cells in *G-Smo* tumors mediate the observed radioresistance. To examine a potential mechanism for radioresistance, we compared OLIG2 phosphorylation in *G-Smo* and *M-Smo* tumors. When phosphorylated, OLIG2 disrupts p53-mediated transcriptional activation[27], which we have shown to be required for radiosensitivity in medulloblastoma[20]. We used western blot to detect phosphorylated OLIG2 (pOLIG2) in lysates of 3 *G-Smo* and 4 *M-Smo* tumors. *G-Smo* tumors consistently showed increased pOLIG2 compared to *M-Smo* tumors (Fig. 3d). These data support a model in which the increased OLIG2+ stem cell populations confer radioresistance in *G-Smo* medulloblastomas, through the mechanism of pOLIG2-mediates inhibition of p53.

**Similar range of cell fates in *G-Smo* and *M-Smo* tumors.** We examined whether the differences in stem cell populations of *G-Smo* and *M-Smo* tumors were accompanied by an expansion of tumor cell fates. We and others have shown that a fraction of medulloblastoma cells trans-differentiate to take on glial phenotypes[10,28]. We previously showed that the 3′ *Yfp* sequence of *SmoM2* can be used as a lineage tracer to identify astrocytes and oligodendrocytes that derive from tumor cells[10]. Comparison of *Yfp* expression in *G-Smo* and *M-Smo* tumors showed that tumor lineage in both genotypes included neural progenitor-like tumor cells, differentiated neurons, astrocytes, and oligodendrocytes (Fig. 4a). Cells within the fibroblast and macrophage/microglia clusters did not express *Yfp*, and the rare *Yfp*+ cells in the endothelial and ependymal clusters did not indicate a significant trend as they were not observed in more than one replicate of either genotype. Glial cells were therefore the only stromal cell types that derived from the tumor lineage in either stem cell-derived or progenitor-derived tumors.

We compared the expression of specific neural markers to assess differences in tumor cell fates. In *G-Smo* tumors, the *hGFAP-Cre* transgene is expected to activate *SmoM2* in a lineage that is broader than the *Atoh1*-lineage activated by *Math1-Cre* in *M-Smo* tumors. The *Atoh1*-lineage comprises rhombic-lip-derived CGNP and unipolar brush cell populations, marked by *Barhl1* and *Eomes*[29] respectively. The set of neuronal cell types with *SmoM2* activation in *G-Smo* tumors includes both rhombic-lip derived populations and ventricular zone-derived GABAergic interneurons and progenitors, marked by expression of *Ascl1*, *Pax3*, and *Pax2*[10]. To determine whether *G-Smo* tumors contained more cells resembling ventricular zone-derived progenitors, we used *Barhl1* and *Eomes* as *Atoh1*-lineage markers and *Ascl1*, *Pax3*, and *Pax2* as ventricular zone-derived interneuron markers, and compared these markers in *G-Smo* and *M-Smo* tumors.

Both genotypes consisted predominantly of *Barhl1*+ cells. We detected a significant genotype-specific difference within the *Atoh1*-lineage, with a higher proportion of *Barhl1*+ cells in *G-Smo* tumors ($p < 1.0 \times 10^{-15}$, two-proportions $z$-test) and a

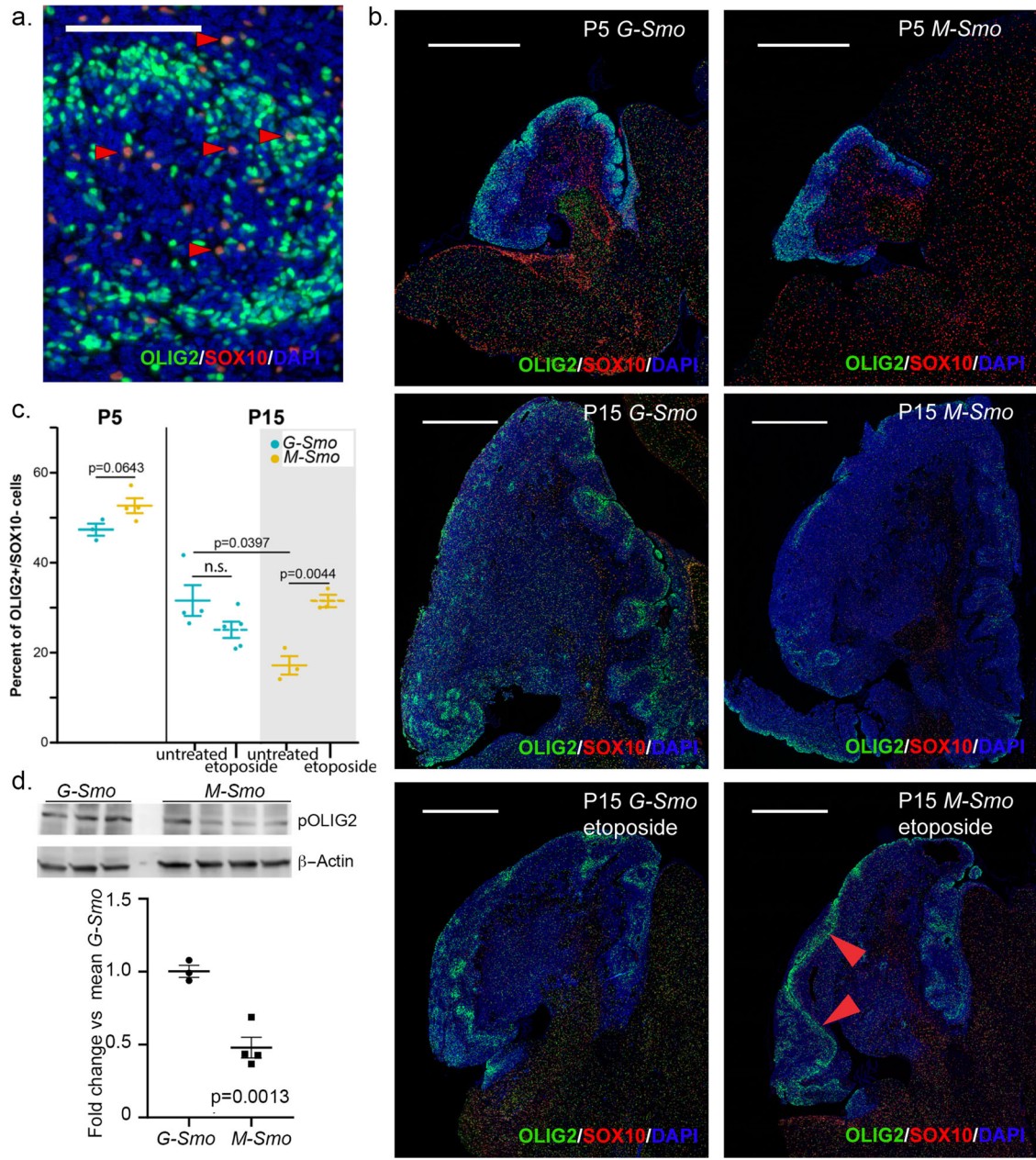

**Fig. 3 OLIG2+ stem cells decline more rapidly in M-Smo compared to G-Smo tumors. a** Representative higher magnification images of OLIG2/SOX10 IHC on a *G-Smo* tumor, showing OLIG2+/SOX10+ oligodendrocytes (arrowheads highlight examples) and OLIG2+/SOX10− tumor stem cells. **b** Representative lower magnification images of OLIG2/SOX10 IHC on sagittal sections from mice of the indicated age, genotype, and treatment. Arrowheads highlight a perivascular region. Scale bars = 100 μm in **a** and 1 mm in **b**. **c** Quantitative comparison of OLIG2+/SOX10− populations in *G-Smo* and *M-Smo* tumors. **d** Western blots of replicate *G-Smo* and *M-Smo* tumors, showing pOLIG2 and β-Actin. Quantification shows pOLIG2 signal, normalized to β-Actin, expressed as fold change relative to mean *G-Smo* value. In graphs in **c** and **d**, Student's *t*-test was used to make pairwise comparisons, dots represent values for individual replicates, bars indicate the means and whiskers indicate the SEM.

higher proportion *Eomes*+ in *M-Smo* tumors ($p < 1.0 \times 10^{-15}$, two-proportions *z*-test) (Fig. 4b). However, we did not detect statistically significant differences in the *Ascl1*+, *Pax*+, and *Pax2*+ GABAergic interneuron-lineage cells derived from the ventricular zone (Fig. 4c). Therefore, although *hGFAP-Cre* activated *SmoM2* in both the rhombic-lip and ventricular zone, cells showing ventricular zone lineage were not expanded in *G-Smo* tumors. We conclude that differences between *G-Smo* and *M-Smo* tumors derive from effects of the timing of oncogenic event on cells that progress through the CGN developmental trajectory, rather than from the recruitment of interneuron-lineage cells for tumor growth.

**Different stromal populations in *M-Smo* and *G-Smo* tumors.** Our scRNA-seq data unexpectedly demonstrated that *M-Smo* and *G-Smo* tumors interact differently with stromal cells in their microenvironments. We compared gene expression in endothelial cell, myeloid cells, and fibroblasts. We selected these cell types because they were the most numerous cell types outside the tumor lineage. To identify tumor-specific changes in these populations, and to distinguish tumor-specific effects common to both tumor genotypes from effects specific to individual tumor genotypes, we combined the scRNA-seq data from *G-Smo* and *M-Smo* tumors with previously obtained data from WT cerebella at P7[10]. We obtained an initial grouping of cells from tumors and

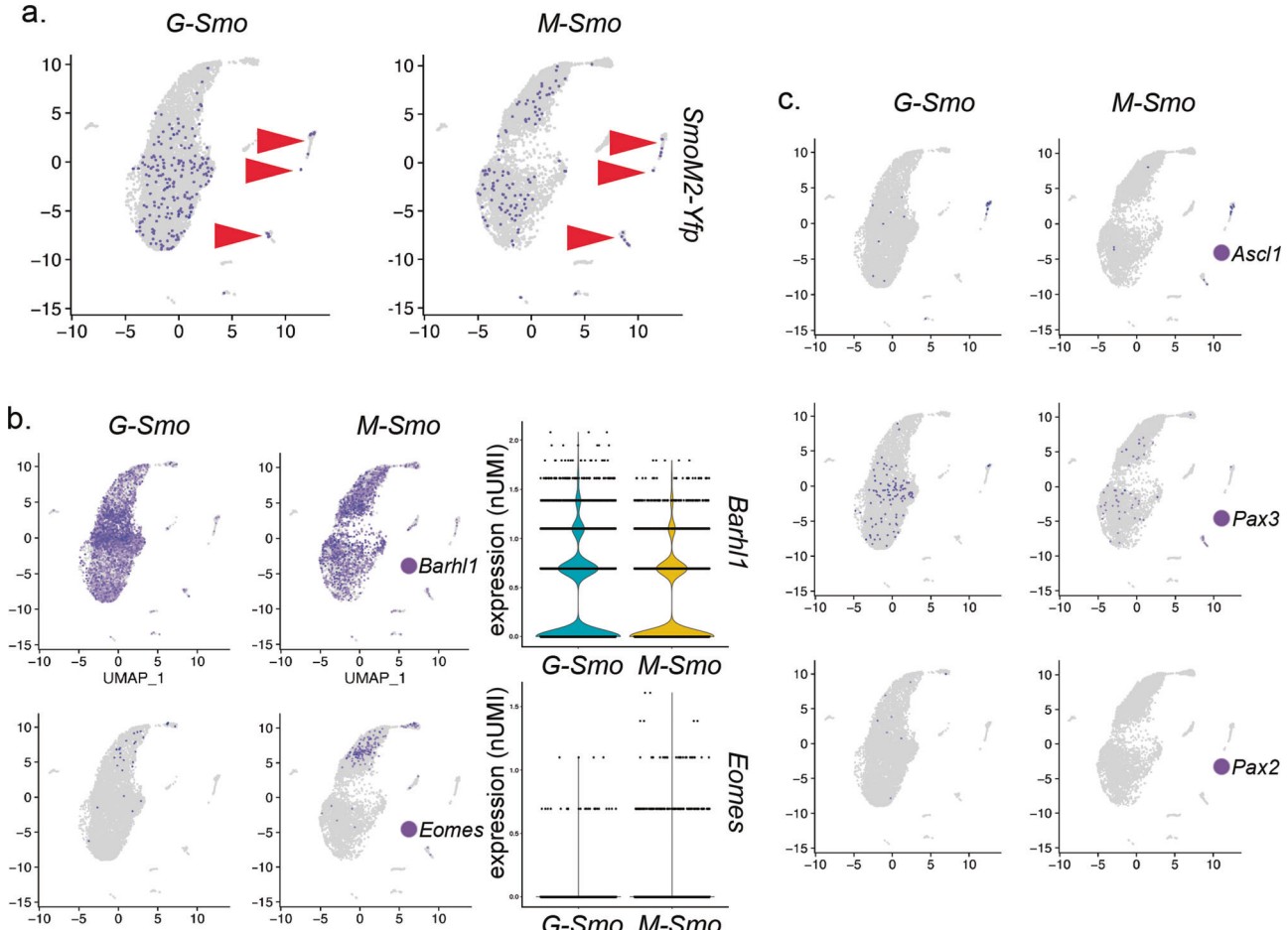

**Fig. 4 Similar range of cell fates in G-Smo and M-Smo tumors.** Feature plots showing expression of the indicated lineage markers, color-coded over the UMAP shown in (2D). **a** SmoM2-Yfp expression, denoting the SmoM2-activated lineage, was distributed similarly in both G-Smo and M-Smo tumors. Arrowheads point to glial clusters. **b** Feature and violin plots show increased Barhl1+ cells in G-Smo tumors and increased Eomes in M-Smo tumors. **c** Feature plots show no significant differences in Ascl1, Pax2, or Pax3 in G-Smo tumors versus M-Smo tumors.

WT mice by cell type using the Harmony algorithm, which co-clustered tumor cells and their most similar normal progenitors[30]. Using Harmony, we generated a UMAP combining G-Smo, M-Smo, and WT cells, color-coded the clusters, and analyzed cluster-specific gene expression profiles; of the proliferative cell types, CGNPs and medulloblastoma cells, grouped together in a set of Barhl1+ clusters, while interneuron progenitors formed a separate group distinguished by Pax3 and Pax2 (Supplementary Fig. 2 and Supplementary Data 6). We identified each stromal cell type based on gene expression (Fig. 5a and Supplementary Data 6), and isolated the endothelial, macrophage/microglial, and fibroblastic populations. We then subjected the cells of each isolated cell type to a new PCA to sub-cluster each cell type.

**Endothelial cells show cancer-specific changes without reflecting developmental differences between tumors.** Endothelial cells showed significant differences between WT and tumor but did not show statistically significant differences between G-Smo and M-Smo tumors. The unsupervised analysis defined 2 clusters (Fig. 5b). Cluster $^E0$ included cells from WT cerebella and both tumor genotype, while $^E1$ was populated predominantly by cells from the tumors, with similar proportions from G-Smo and M-Smo (Fig. 5c, d). Both $^E0$ and $^E1$ expressed the endothelial markers Pecam1 and Cldn5, confirming endothelial identity (Fig. 5e).

Each cluster showed cluster-specific gene expression (Fig. 5f, g and Supplementary Data 7). The tumor-specific $^E1$ cells, showed increased expression of genes likely to contribute to malignancy, including the VEGF receptor Flt1, the p-Glycoprotein Abcb1a (aka Mdr1), and the CXCR4 ligand Cxcl12 (aka Sdf1), which has been shown to promote medulloblastoma growth and glioblastoma–endothelial interactions[31–34]. Biologically relevant differences between endothelial populations in WT cerebella and tumors were shared between tumor genotypes, consistent with their overall similarity.

**Developmental differences alter tumor-associated myeloid populations.** The cells with myeloid characteristics, in contrast to endothelial cells, differed significantly between M-Smo and G-Smo tumors. Unsupervised analysis grouped the myeloid-like cells into 5 clusters, $^M0$–$^M4$ (Fig. 6a). Projection of C1qb expression confirmed that clusters $^M0$–$^M3$ were populated by myeloid cells (Fig. 6b). In contrast, cluster $^M4$, which was the least populated, was C1qb- and expressed Cnn3 and Meis1 (Fig. 6b); this marker pattern identified cluster $^M4$ as choroid plexus epithelial cells that have been noted to cluster with myeloid cells in other scRNA-seq analyses[35].

We identified the types of myeloid cells in each cluster by defining the sets of genes upregulated by cells within the cluster, compared to cells in the other 4 clusters (Supplementary Data 8).

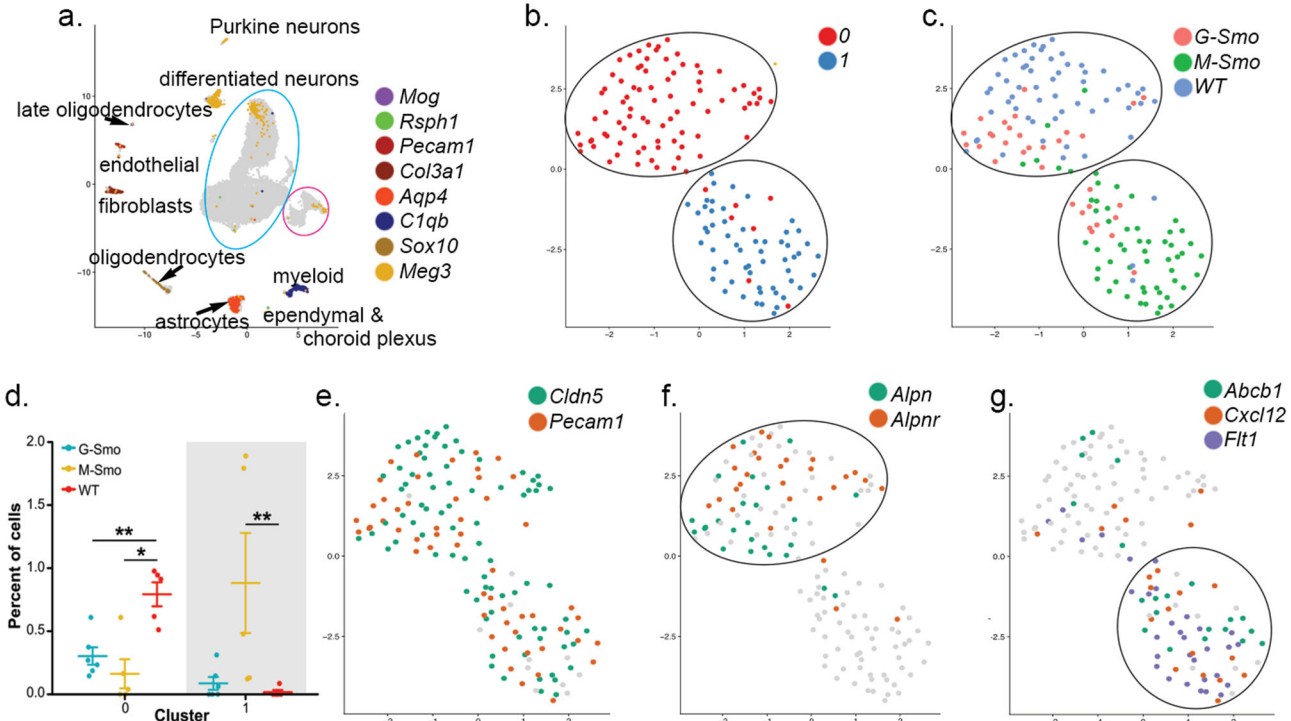

**Fig. 5 Tumor-specific gene expression in endothelial cells. a** UMAP plot of cells from *G-Smo* and *M-Smo* tumors combined with cells from P7 WT cerebella, analyzed using the Harmony algorithm. The blue circle shows the region where neural progenitor-like tumor cells and CGNPs localize. The pink circle shows the GABAergic interneuron lineage. Stromal cell types are identified by the expression of the indicated, color-coded markers. **b** UMAP showing endothelial cells from *G-Smo*, *M-Smo*, and P7 WT, with clusters color-coded. **c** genotype color-coded on the UMAP from **b**. Circles indicate the positions of the clusters. **d** Comparison of the indicated cluster populations in *G-Smo*, *M-Smo*, and P7 WT. Student's test was used in **d**. Dots represent values for individual replicates, bars indicate the means and whiskers indicate the SEM. **e–g** Feature plots on UMAP from **b**, showing expression of **e** endothelial markers *Cldn5* and *Pecam1*, **f** Cluster $^E0$ markers *Apln* and *Aplnr*, or **g** tumor-specific Cluster $^E1$ markers *Abcb1a* (aka *Mdr1*), *Cxcl12*, and *Flt1*.

We projected key cluster-specific marker genes on a UMAP of the 4 clusters, along with other known markers of myeloid phenotype (Fig. 6c–f). Clusters $^M0$ and $^M1$, which localized to clusters on one side of the UMAP, expressed *Cx3cr1* which distinguished them as microglia, while clusters $^M2$ and $^M3$, opposite in the UMAP, showed minimal *Cx3cr1*, indicating that they were macrophages (Fig. 6c). $^M0$ cells showed cluster-specific expression of *Sparc* (Fig. 6c), identifying them as mature, ramified microglia[36]. $^M1$ microglia specifically expressed *Mrc1* and *Wfdc17* (Fig. 6d) which have been linked to an M2-like, anti-inflammatory phenotype[37–39]. Both $^M0$ and $^M1$ cells expressed *Igf1* (Fig. 6d), which has been shown to be a paracrine signal promoting growth in SHH medulloblastoma[28]; *Igf1* was not detected in $^M2$ and $^M3$ cells.

$^M2$ macrophages specifically expressed MHCII components, including Histocompatibility 2, Class II Antigen E alpha (*H2-Ea*), and the Invariant Polypeptide of Major Histocompatibility Complex, Class II Antigen-associated (*Cd74*) as well as *Il1b* and *Ccr2* (Fig. 6e), consistent with a pro-inflammatory M1-like phenotype[40–42]. Importantly, *Ccr2+* macrophages have previously been shown to exert an anti-tumor effect in a medulloblastoma[42]. $^M3$ macrophages specifically expressed *Cd163* and *Mrc1* (Fig. 6f), also consistent with an anti-inflammatory M2 phenotype[43,44]. Myeloid cells thus resolved into microglial and macrophage populations, each with M1-like and M2-like subsets (Table 2). $^M0$ and $^M1$ cells expressed *Igf1*, which has been shown to promote medulloblastoma progression, while $^M2$ expressed *Ccr2*, which is associated with tumor-inhibiting macrophages.

Each cluster distributed differently across *G-Smo* and *M-Smo* tumors and WT cerebella (Fig. 6g, h). The M1-like microglial

cluster $^M0$ and the non-myeloid cluster $^M4$ were distributed relatively evenly in all 3 genotypes. *M-Smo* tumors included M1-like and M2-like microglia (clusters $^M0$ and $^M1$), and M1-like and M2-like macrophages (cluster $^M2$ and $^M3$). In contrast, the myeloid populations of *G-Smo* tumors were more limited, with significantly fewer $^M2$ and $^M3$ cells. Consistent with reduced *Igf1*-$^M2$ and $^M3$ populations, *G-Smo* tumors contained larger *Igf1+* fractions of myeloid cells compared to *M-Smo* tumors ($p = 0.012$; *t*-test).

**Differential cytokine expression in *G-Smo* and *M-Smo* tumors.** To consider potential mechanisms for the differences in macrophage populations in the two tumor types, we performed a non-biased comparison of cytokine expression. To generate a list of known cytokines and chemokines, we used the set of 232 genes tagged with the Gene Ontology term "Cytokine Activity". For each gene, we conducted differential expression testing between *G-Smo* and *M-Smo* tumors. Macrophage Migration Inhibitory Factor *(Mif)* was the only differentially expressed cytokine, and was higher in *G-Smo* tumors compared to *M-Smo* tumors (Fig. 6i)[45]. MIF is a ligand for CD74[45], and increased *Mif* in *G-Smo* tumors may contribute to the markedly lower *Cd74+* population. Prior studies in human glioblastoma and melanoma associate MIF with cancer stem cells show that intercellular communication through MIF-CD74 is immunosuppressive, and that blocking MIF-CD74 signaling increases tumor-associated M1 macrophages[46–48]. Based on these prior studies and the inverse correlation of MIF and CD74 in our tumor model, we propose that MIF functions in medulloblastoma to bias myeloid cells toward an *Igf1+* phenotype, and acts more effectively in *G-Smo* tumors, which have higher *Mif* expression.

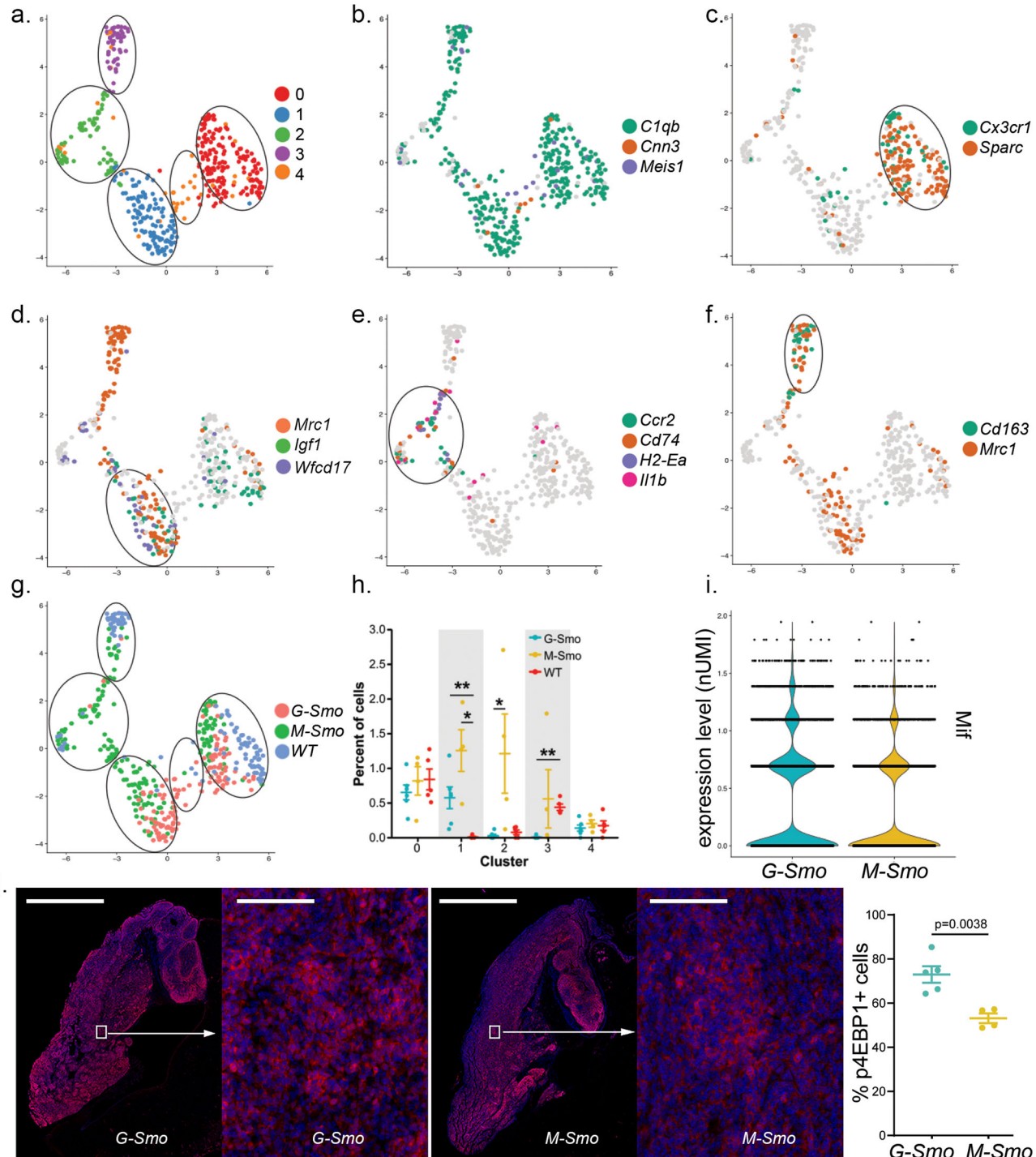

**Fig. 6 M-Smo tumors show macrophage and microglial populations not present in G-Smo tumors. a** UMAP showing myeloid-like cells from *G-Smo*, *M-Smo,* and P7 WT, with clusters color-coded and circled. **b** Feature plot on UMAP from **a**, showing expression of the pan-macrophage and microglial marker *C1qb* and the choroid plexus epithelial markers *Cnn3* and *Meis1*. Cluster 4 does not show *C1qb* expression. **c**–**f** Feature plots on UMAP from **a**, showing expression of **c** Cluster [M]0 markers *Cx3cr1* and *Sparc*, **d** Cluster [M]1 markers *Igf1, Mrc1,* and *wfdc17*, or **e** Cluster [M]2 markers *Ccr2, Cd74, H2-Ea-ps*, and *Il1b* **f** Cluster [M]3 markers *Mrc1* and *Cd163*. **g** genotype color-coded on the UMAP from **a**. Circles indicate the positions of the clusters. **h** Comparison of the indicated cluster populations in *G-Smo*, *M-Smo*, and P7 WT. **i** Violin plots showing the number of cells expressing the indicated level of *Mif* in each genotype. **j** IHC for p4EBP1 in representative *G-Smo* and *M-Smo* tumors at low and higher magnification, and quantification of p4EBP1+ fractions of tumor cells in each genotype. Scale bars = 2 mm in low magnification images and 200 μm in higher magnification images. Student's test was used in **h**, **j** and two-proportions z-test in **i**. In graphs in **h** and **j**, dots represent values for individual replicates, bars indicate the means and whiskers indicate the SEM.

**Table 2 Identification of clusters with myeloid characteristics ($^M0$–$^M4$).**

| Cluster | Cell-type designation | Distinctive markers |
|---|---|---|
| $^M0$ | Mature microglia | Igf1, Cx3cr1, Sparc |
| $^M1$ | M2 microglia | Igf1, Wfdc17, Mrc1 |
| $^M2$ | M1 macrophages | Ccr2, Cd74, H2-Ea, Il1b |
| $^M3$ | M2 macrophages | Cd163, Mrc1 |
| $^M4$ | Choroid plexus epithelial cells | Cnn3, Meis1 |

**Differential mTORC1 activation in *G-Smo* and *M-Smo* tumors**. To determine if this difference in *Igf1* may be biologically significant, we analyzed mTORC1 activity, which is increased by IGF1 signaling and known to be important in SHH medulloblastoma progression[49,50]. We measured mTORC1 activity by quantifying cells showing phosphorylated 4EBP1 protein in 5 *G-Smo* and 4 *M-Smo* tumors, using IHC. Both genotypes showed abundant p4EBP1+ cells in tumors, with significantly higher fractions p4EBP1+ cells in *G-Smo* tumors (Fig. 6j). The increased fraction of *G-Smo* cells showing mTORC1 activation is consistent with increased paracrine stimulation via IGF1. The difference in *Igf1*-expressing myeloid cells in *G-Smo* and *M-Smo* tumors demonstrates an effect of tumor genotype on the TME that may feedback on tumor phenotype by altering tumor mTORC1 activation, and thus contribute to differences in tumor growth and recurrence.

**Differential myeloid marker expression in mouse and human SHH medulloblastomas**. We confirmed differential myeloid populations using IHC for the MHCII glycoprotein coded by *H2-Ea* and the pan macrophage/microglial marker IBA1, comparing 4 *G-Smo* and 3 *M-Smo* tumors (Fig. 7a). H2-EA+ macrophages were more prevalent in *M-Smo* tumors (Fig. 7b), consistent with the scRNA-seq data. These results confirm that the protein expression of these markers matched the transcript data from scRNA-seq and demonstrate that immunohistochemical staining for MHCII components was sufficient to distinguish between *G-Smo* and *M-Smo* tumors.

To determine whether similar differences in myeloid markers can be used to probe clinical medulloblastoma samples, we used IHC to detect HLA-DR proteins, the human orthologs of mouse H2-EA, as described previously[51]. We analyzed 30 medulloblastoma samples for which SHH, WNT, group 3, and group 4 subgroups and subtypes within each subgroup had been using bulk transcriptomic and methylomic analysis according to published criteria[2,52]. All slides were subjected to blinded review and the frequency of HLA-DR+ cells was scored on a scale of 0–3 (Fig. 7c)[51]. 10 of the medulloblastomas were SHH-subgroup tumors, and these tumors received significantly higher HLA-DR scores compared to tumors of the other types (Fig. 7d). Subtype was determinable for 9/10 SHH medulloblastomas, including 2 SHH-alpha tumors, 4 SHH beta tumors, and 3 SHH delta tumors. HLA-DR staining varied between SHH subtypes, with SHH-alpha tumors showing markedly less HLA-DR compared to beta and delta tumors (Fig. 7e). SHH beta and SHH delta are the infant-predominant subtypes[2], and the trend toward higher HLA-DR expression in these subtypes suggests that a developmental process affects myeloid subgroups in human tumors, as in our mouse models.

The trends noted in our IHC analysis correlate well with bulk transcriptomic data from prior studies. In the published data first used to establish the subtypes within the 4 medulloblastoma subgroups[2], SHH-subgroup tumors showed significantly higher *HLA-DRA* mRNA (Fig. 7f), and SHH-alpha subtype tumors showed significantly lower *HLA-DRA* mRNA, compared to the other SHH subtypes (Fig. 7g). We also examined SHH-subgroup tumors stratified by age, irrespective of subtype; infant medulloblastomas showed significantly higher *HLA-DRA* (Fig. 7h), consistent with developmental differences producing differences in the TME, as in our model.

**Developmental differences influence tumor fibroblast populations**. Genotype-specific differences in the TME were not limited to myeloid cells; we also noted differences in the fibroblast populations. Fibroblasts in *G-Smo* and *M-Smo* tumors and P7 WT cerebella assorted into 3 clusters, with significant differences in the types of fibroblasts in each genotype (Fig. 8a–c). Each cluster showed specific gene expression patterns (Fig. 8d–f and Supplementary Data 9), with genotype-specific effects retinoid signaling genes; *Fabp5* was WT-specific, and *Rbp4* and *Crabp2* were specifically down-regulated in *G-Smo* tumors (Fig. 8g). These differential patterns show that the timing of oncogenesis affects the fibroblastic stroma of the TME as well as the tumor-associated myeloid cells.

**Discussion**

Our data show that medulloblastomas initiated by a common driver mutation at different points in a developmental trajectory can show similar transcriptomic profiles but contain significantly different populations of tumor and stromal cells and respond differently to therapy. *G-Smo* and *M-Smo* tumors both showed transcriptomic profiles in microarray studies that were consistent with SHH-subgroup medulloblastoma. However, scRNA-seq analysis demonstrated that *G-Smo* tumors, generated by initiating *SmoM2* expression in *Gfap*-expressing CNS stem cells, contained more proliferating cells at P15 and more *Olig2*+ tumor stem cells. In contrast, *M-Smo* tumors, generated by initiating *SmoM2* expression in *Atoh1*-expressing committed neural progenitors, contained more differentiating tumor cells at P15.

Our analysis of *Olig2*+ stem cell populations showed that the temporal pattern of stem cell regulation was different in *M-Smo* and *G-Smo* tumors. In the early stages of tumor growth at P5, both *G-Smo* and *M-Smo* tumors consisted mostly of OLIG2+ stem cells. As tumors grew over time, the fraction of OLIG2+ cells decreased in both genotypes, but *G-Smo* tumors retained larger OLIG2+ stem cell populations. The proliferation rate in the OLIG2+ populations was not different at P15, indicating that the difference in OLIG2+ population size is due to differences in the tendency of OLIG2+ cells or their progeny to maintain the stem cell phenotype. The developmental history of the tumors thus continued to influence tumor stem cell behavior weeks after tumor initiation.

Developmental history also influenced the interactions of *G-Smo* and *M-Smo* tumors with the TME. Not all stromal cell types showed strong differences between *G-Smo* and *M-Smo* tumors. Both tumor genotypes showed endothelial populations that were configured to support malignancy, with increased expression of the tumor stimulating cytokine *Cxcl12* and the drug efflux pump *Abcb1* (aka *Mdr1*), compared to endothelial cells in WT cerebella. However, these tumor-specific patterns of gene expression were not significantly different in *G-Smo* and *M-Smo* tumors. In contrast, myeloid and fibroblastic cells showed differences both between tumor versus WT and *G-Smo* versus *M-Smo* tumors, with *M-Smo* tumors harboring more anti-tumor *Cc2*+ cells and fewer tumor-promoting *Igf1*+ cells. Different patterns of cytokine expression, with increased *Mif* expression in *G-Smo* tumors, may mediate the different TME interactions. As the tumor-TME interactions include oncogenic, paracrine signaling, these differences in tumor and stromal populations produce complex, re-

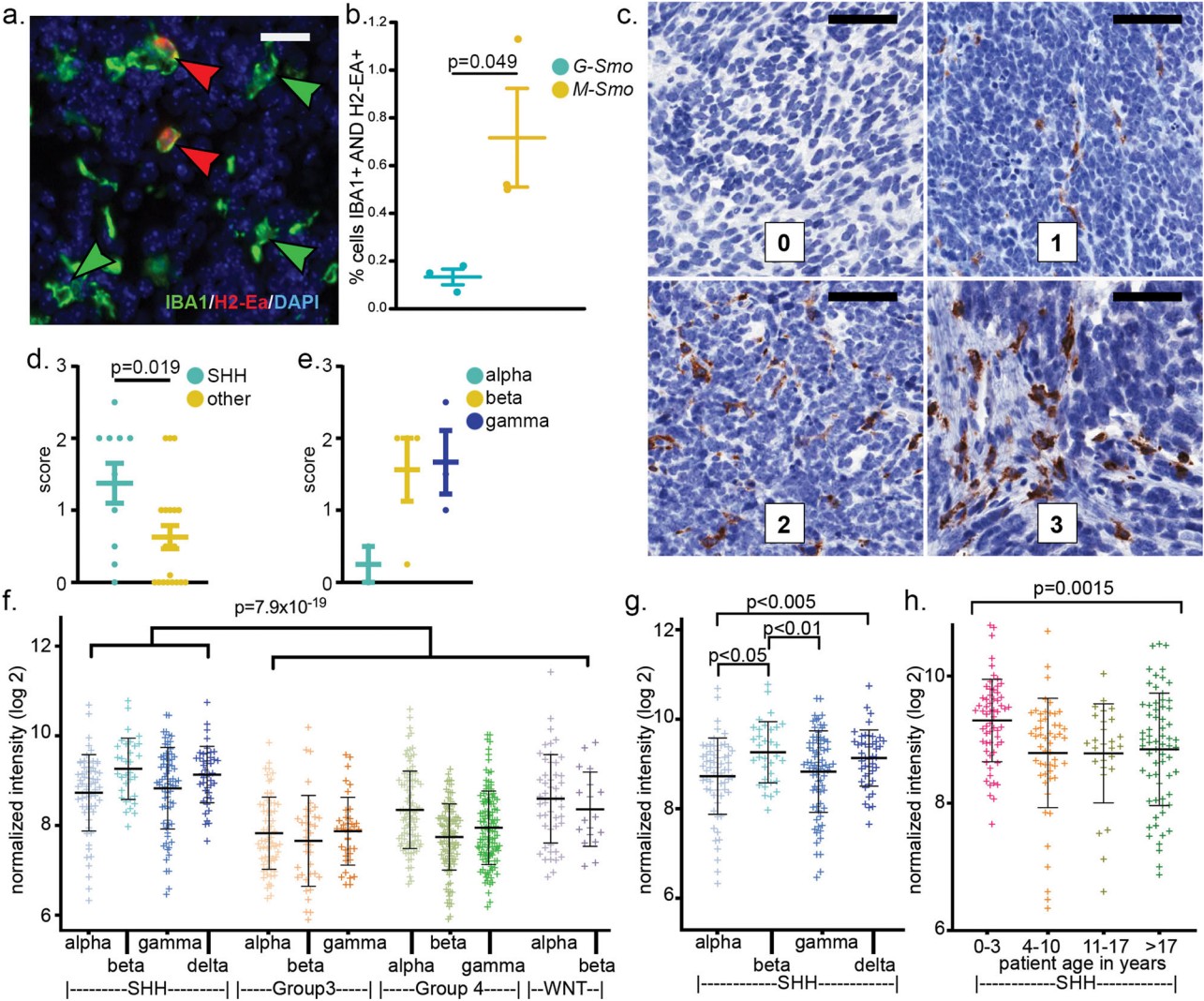

**Fig. 7 Differential MHCII/HLA expression in G-Smo versus M-Smo tumors and in human medulloblastoma subsets. a** IHC for IBA1 and H2-Ea in a representative M-Smo tumor. **b** Quantification of the fraction of IBA1+/H2-Ea+ cells *G-Smo* and *M-Smo* tumors. **c** Representative IHC in human medulloblastomas using a pan-HLA-DR antibody, with scores as noted. **d**, **e** Quantification of the HLA-DR scores of human medulloblastomas for the indicated tumor types. **f**–**h** Analysis of *HLA-DRA* mRNA in published human medulloblastomas microarray data, comparing **f** SHH-subgroup versus all other subgroups, **g** SHH-alpha versus each other SHH subtype, or **h** SHH-subgroup stratified by age. Scale bars = 20 µm in **a** and 50 µm in **c**. Student's test was used in **b**, **d**, **e** and one-way ANOVA was used in **f**–**h**. In all graphs, points represent values for individual replicates, bars indicate the means and whiskers indicate the SEM.

enforcing effects; thus, differences in *Igf1*-expressing myeloid populations correlate with differences in mTORC1 activation in tumor cells Differences in developmental history, therefore did not produce large differences in global gene expression profile, but significantly altered the heterogeneity within tumors, producing differences that can be self-amplifying.

Our analysis identified multiple processes that are likely to contribute mechanistically to the differences in the progression and recurrence of *G-Smo* and *M-Smo* tumors. *Olig2*-expressing tumor stem cells, *Ccr2*+ macrophages, and *Igf1*+ microglial have all been shown to affect tumor progression and prognosis[25,28,42]. The importance of *Olig2*+ medulloblastoma stem cells was shown in prior studies where targeting the *Olig2*+ population, either by conditional ablation of *Olig2*-expressing cells using HSV TK or conditional genetic deletion of the *Olig2* locus, reduced the growth of SHH-driven medulloblastomas in mice[25]. We propose that by preventing p53-dependent apoptosis after xRT, pOLIG2 in tumor stem cells may allow survival long enough for DNA repair, as seen in *Bax*-deleted medulloblastomas after xRT[20].

Differences in *Igf1*+ and *Ccr2*+ myeloid cells may also contribute to poor outcomes in *G-Smo* mice. *Ccr2*-expressing macrophages, which are reduced in *G-Smo* tumors, have been shown to suppress tumor growth[42]. In contrast, *Igf1*-secreting microglia, which are increased in *G-Smo* tumors, support medulloblastoma progression[28]. The differences in the polarization of myeloid cells in *G-Smo* and M-Smo tumors, along with differences in *Olig2*+ stem cells, are multiple mechanisms that are each sufficient to worsen prognosis, and these mechanisms may act in combination.

Several factors may obscure important features in bulk transcriptomic studies. Transcriptomic signals from small subsets of cells may not be detectable when averaged with larger subsets of cells in bulk tumor lysates. For example, *Ccr2*+ cells did not produce a detectable signal in our microarray study but were detectable by scRNA-seq. In addition, cell types that are very common, such as proliferating cells, may generate large signals that do not produce statistically detectable variation, as we found that proliferation markers were not statistically different in bulk

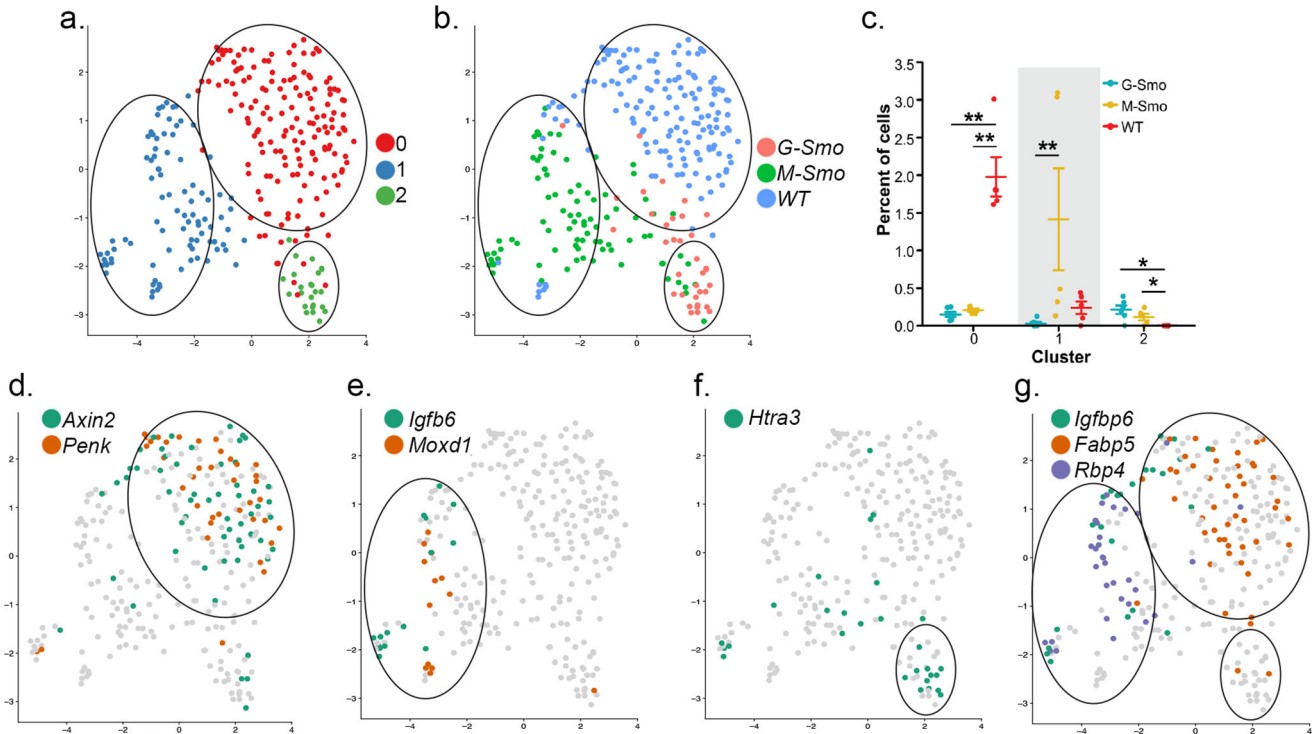

**Fig. 8 Different fibroblastic populations in G-Smo and M-Smo tumors. a** UMAP showing fibroblastic cells from *G-Smo*, *M-Smo*, and P7 WT, with clusters color-coded and circled. **b** genotype color-coded on the UMAP from **a**. Circles indicate the positions of the clusters. **c** Comparison of the indicated cluster populations in *G-Smo*, *M-Smo*, and P7 WT. Dots represent values for individual replicates, bars indicate the means and whiskers indicate the SEM. **d–g** Feature plots on UMAP from **a**, showing expression of **d** Cluster [F]0 markers *Axin2* and *Penk*, **e** Cluster [F]1 markers *Igfbp6* and *Moxd1*, **f** Cluster [F]2 marker *Htra3*, and **g** retinoic acid signaling components *Crabp2*, *Fabp5,* and *Rbp4*. Student's test was used in **c**.

transcriptomic analysis but proliferative cells were enriched in *G-Smo* tumors. Additionally, genes that are important markers within one cell type may be expressed by other cell types that are also present in the samples. For example, *Olig2* is expressed in medulloblastoma samples by tumor stem cells and by oligodendrocytes. Bulk transcriptomic datasets cannot distinguish *Olig2* mRNA expressed by oligodendrocytes from stem cell-derived *Olig2*. In contrast, studies with cellular resolution, including scRNA-seq and IHC, allow the detection of both small populations, the counting of proliferative cells identified by multiple markers, and the differentiation of markers by their cellular context. While bulk transcriptomic analysis of *HLA-DRA* mRNA showed trends that matched our HLA-DR IHC study, the variability between samples in each subtype of the bulk analysis was large; statistical trends could be discerned between subtypes, but each subtype included individuals within a wide range of values. Larger studies are needed to determine if cell-resolved data for markers, including IHC for HLA-DR or OLIG2, may allow stratification into more homogeneous groups than the SHH subtypes identified by bulk methods.

The developmental differences between *G-Smo* and *M-Smo* tumors are complex and include differences in both timing and lineage. *Gfap* expression, and thus *SmoM2* activation in *G-Smo* tumors, begins earlier than *Atoh1* expression and *SmoM2* activation in *M-Smo* tumors. Moreover, the *hGFAP-Cre* transgene is likely to activate *SmoM2* in Nes+ /Atoh1– neural stem cells in the EGL; these cells are known to be SHH sensitive but do not express *Atoh1* and thus are likely not to be targeted by *Math1-Cre*. As a result, *G-Smo* tumors initiate in both *Atoh1*+ and *Nes*+ /*Atoh1*– lineages, while *M-Smo* tumors initiate in only the *Atoh1*+ lineage. However, we have previously shown that *SmoM2* activation relaxes lineage restriction in *M-Smo* tumors, increasing the tendency of *Atoh1*-lineage cells to take on glial fates[10]. The lineage

and timing differences are thus interlinked and may be impossible to deconvolute. The net result, however, is to produce consistent differences in tumor heterogeneity.

Our data show that developmental events that would be cryptic in a clinical setting can influence the clinical outcomes by affecting both tumor cells and stroma. While these effects passed undetected in bulk transcriptomic studies, the fractions of OLIG2+ tumor cells and MHCII-expressing macrophages were effective biomarkers that distinguished radioresistant stem cell-derived tumors from radiosensitive tumors originating in CGNPs. These biomarkers succeeded because they were sensitive to differences in cellular heterogeneity. Similarly, *Olig2*+ populations and myeloid subtypes are readily measurable in clinical samples. While scRNA-seq identified differences between tumors, each of these differences can also be probed in clinical samples using IHC or flow cytometry, which preserves cellular information that is lost in transcriptomic analysis. Future studies are needed to determine whether analysis of these parameters provides prognostic information that reduces the heterogeneity of transcriptomic subgroups, improving prognostication and precision therapy.

## Methods

**Mice**. We crossed *SmoM2* mice (Jackson Labs, stock # 005130) with *hGFAP-Cre* mice (Jackson Labs, stock # 004600), to generate *G-Smo* mice, and with *Math1-Cre* (Jackson Labs, stock #011104) to generate *M-Smo* mice. All mice were of species *Mus musculus* and crossed into the C57BL/6 background through at least five generations. All animal studies were carried out with the approval of the University of North Carolina Institutional Animal Care and Use Committee under protocols (19-098).

**Human subjects**. Medulloblastoma samples were obtained from patients consented under the University of Colorado IRB COMIRB 95-500. All patient materials were de-identified prior to their use in this study.

**Histology, IHC, and western blot**. For histology and IHC, mouse brains were processed, immunostained, and quantitatively analyzed as previously described[10,20,53]. For western blot, samples were snap-frozen and lysed, blotted, and stained as previously described[53]. Primary antibodies used were: cC3 1:100 (Biocare, #229), pRB diluted x (x), OLIG2 diluted 1:100 (Cell Marque, # 387R-14), pOLIG2 diluted 1:3000 (x), SOX10 diluted 1:200 (Cell Signaling Technology, #7833S), MHC Class II glycoprotein H2-Ea diluted 1:200 (Novus, # NBP1-43312), IBA1 1:2000 (Wako Chemicals, #019-19741) and HLA-DR diluted 1:50 (Wako NBP2-47670), and p4EBP1 diluted 1:500. HLA-DR studies were performed as previously described[51]. All other stained slides were counterstained with DAPI, digitally imaged using an Aperio Scan Scope XT (Aperio), and subjected to automated cell counting using Tissue Studio (Definiens).

**Radiation therapy and survival studies**. Medulloblastoma-bearing mice were treated with 10 Gy X-ray irradiation, delivered as 5 fractions of 2 Gy each, as previously described[20]. Briefly, starting P10, G-Smo, and M-Smo mice were irradiated daily for 5 days. Irradiation was performed under general anesthesia with isoflurane, delivered by vaporizer through nose cones, after which mice were allowed to recover and then returned to their dams. Following radiation therapy, radiation-treated mice and untreated littermate controls were observed for symptoms of tumor progression, including movement disorder, ataxia, or sustained weight loss. Mice showing symptoms of progression were euthanized, and the time to progression was considered to be the EFS. Brain pathology was analyzed to confirm tumor progression.

**Microarray analysis**. Medulloblastomas were harvested from P15 G-Smo and M-Smo mice and immediately flash frozen. Some P15 G-Smo and M-Smo mice were subjected to 10 Gy xRT as a single fraction 2 h before harvest. Frozen tissue was homogenized by sonication in RLT buffer (Qiagen), and total RNA was purified following the manufacturer's instructions (QIAGEN, cat#74104). RNA was labeled, hybridized to Affymetrix Mouse Gene 2.1ST arrays, per manufacturer's protocol (Affymetrix, Santa Clara, CA USA), and scanned by the UNC-Lineberger Genomics core. Microarray analysis was performed using the Partek Genomics Suite (Partek Incorporated, St. Louis, Missouri). 1-way ANOVA was used to identify genes that varied significantly between the two genotypes and 2-way ANOVA was used to analyze variation in xRT-treated and untreated G-Smo and M-Smo tumors.

**Tissue preparation for Drop-seq**. Mice were anesthetized using isoflurane and then euthanized via decapitation. The brain was divided along the sagittal midline and one half was processed for histology while a large sample of tumor was dissected from the other half and processed for Drop-seq analysis. This sample was dissociated using the Papain Dissociation System (Worthington Biochemical) following the protocol used in previous studies[10,54]. Briefly, tumor samples were incubated in papain at 37 °C for 15 min, then triturated, and the suspended cells were spun through a density gradient of ovomucoid inhibitor.

Pelleted cells were then resuspended in 1 mL HBSS with 6 g/L glucose and diluted in PBS-BSA solution to a concentration of 95–110 cells/µL. Barcoded Seq B Drop-seq beads (ChemGenes) were diluted in Drop-seq lysis buffer to a concentration between 95–110 beads/µL. Tumor cells were co-encapsulated with barcoded beads using FlowJEM brand PDMS devices as previously described[10]. All cells were processed within one hour of tissue dissociation. Droplet breakage and library preparation steps followed Drop-seq protocol V3.1[55]. After PCR, amplified cDNA was subjected Ampure XP cleanup at 0.6× and 1× ratios to eliminate residual PCR primers and debris. found by the bioanalyzer electropherogram. 1.If PCR failed to generate adequate cDNA, the PCR was repeated with the 3rd round increased from 11 to 13 cycles.

For QC purposes, library pools consisting of the tagmented cDNA from 2000 beads/run were prepared and sequenced to low depth (~2.5 M reads/2 K beads). We used the resulting data to assess library efficiency, including total read losses to PolyA regions, nonsense barcodes, and adapter sequences as well as the quality and number of the transcriptomes captured. Passable runs contained 40–60% of reads associated with the top 80–100 barcodes found in 2000 beads.

Drop-seq runs passing QC were then prepared for high-depth sequencing on an Illumina Hi-Seq 4000. Each sample underwent a new generation of bulk cDNA from the stored beads and was prepared with the same ratios as described above. Pools were formulated according to the number of cells/sample to avoid oversampling of each sample and to balance the reads per lane across the Hi-Seq.

**Processing of scRNA-seq data**. Full data analysis code is available at https://github.com/malawsky/Gershon_single-cell.

Data analysis was performed using the Seurat R package version 3.1.1[56]. Data were subjected to several filtering steps. First, only genes that were detected in at least 30 cells were considered, to prevent misaligned reads appearing as rare transcripts in the data. Cells were then filtered using specific QC criteria to limit the analysis to cells with transcriptomes that were well-characterized and not apoptotic.

We noted that G-Smo cells were sequenced at a greater depth than M-Smo cells which can introduce unwanted batch effects into the analysis. Consistent with best practices[24], we downsampled the G-Smo cells to 60% of their original depth so as to achieve similar sequencing depth between G-Smo and M-Smo cells prior to further filtering.

Putative cells with fewer than 500 detected RNA molecules (nCount) or 200 different genes (nFeature) were considered to have too little information to be useful, and potentially to contain mostly ambient mRNA reads. Putative cells with >4 standard deviations above the median nCount or nFeature were suspected to be doublets, improperly merged barcodes, or sequencing artifacts and were excluded. As in our previously published work, putative cells with more than 10% mitochondrial transcripts were suspected to be dying cells and also excluded[10].

In total, 53% of putative cells from G-Smo mice and 49% of putative cells from M-Smo mice met QC criteria and were included in the analysis. From the 6 G-Smo mice, we included a total of 8699 cells with a range of 802–2056 cells per animal and a median of 1481 cells. From the five M-Smo mice, we included a total of 5930 cells, with a range of 614–2512 cells per animal and a median of 821 cells.

**scRNA-seq Data normalization, clustering, differential gene expression, and visualization**. The data were normalized using the SCTransform method as implemented in Seurat. The function then selected the top 3000 most highly variable genes. PCA was performed on the subset of highly variable genes using the RunPCA function. The number of PCs to be used in the downstream analysis was chosen to be 17 based on examining the elbow in the elbow plot as implemented by Seurat.

We used the FindNeighbors and FindClusters functions to identify cell clusters in the data. Briefly, these functions define a graph connecting cells to each other by weighted edges and then identify clusters in the graph that place each cell into a single cluster using the Louvain algorithm. For the FindClusters function, we found that a resolution of 1.2 produced biologically meaningful clusters.

To identify differential genes between clusters of cells, Wilcoxon rank-sum test was used to compare gene expression of cells within the cluster of interest to all cells outside that cluster as implemented by the FindMarkers function. Specific parameters for the genes to be analyzed based on their log fold change between the two compared groups and percent of cells expressing the gene in at least one of the groups are available in the data analysis code. Uniform Manifold Approximation and Projection were used to reduce the PCs to two dimensions for data visualization using the RunUMAP function. For re-iterated analysis of the isolated stromal clusters, the same procedures were used with parameters changed as described in the data analysis code.

**Cell-type identification**. Following PCA and UMAP, we inspected clusters for expression of indicated markers using the differential gene expression results. Marker genes were plotted using an expression cut off to facilitate the visualization of both high- and low-expression genes on a single plot. Cutoffs are applied so that only cells with expression >cutoff received the color corresponding to that gene. These cutoffs are available in the data analysis code. In feature plots of multiple genes, for individual cells expressing multiple markers, each gene was over-plotted in the order described in the code. Feature plots of genes will be made available upon reasonable request.

**Harmony analysis**. To merge the previously published WT data set with the tumor data set, we used the Harmony algorithm[30]. First, the WT and tumor data set were analyzed in single SCTransform normalization and PCA steps. The Harmony algorithm then used the cells' PCA coordinates and data set identity to calculate new coordinates for each cell so as to minimize data set dependence when applying clustering to the cells. This algorithm produced a dimensional reduction that was used in place of PCA with the same steps applied to the data as described in the "Data normalization, clustering, differential gene expression, and visualization" and "Cell-type identification" subsections of the "Methods" section.

**Reporting summary**. Further information on research design is available in the Nature Research Reporting Summary linked to this article.

## Data availability
The microarray data were deposited Gene Expression Omnibus database under the accession code GSE155471. The scRNA-seq data were deposited in the Gene Expression Omnibus database under the accession code GSE150579. Expression data is also available through our web-based application https://gsmovmsmovieworer.shinyapps.io/GvMviewer/.

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

## Acknowledgements

We thank Dr. David Rowitch and Dr. Robert Wechsler-Reya for sharing their invaluable editorial and scientific insight during the writing process. We thank the UNC CGBID Histology Core, supported by P30 DK 034987, and the UNC Tissue Pathology Laboratory Core supported by NCI CA016086. J.O. was supported by NINDS (F31NS100489). T.D. was supported by NINDS (R01NS102627, F31NS120459). T.R.G. was supported by NINDS (R01NS088219, R01NS102627, R01NS106227) and by the UNC Department of Neurology Research Fund. T.R.G., K.W., and B.B. were supported by a TTSA grant from the NCTRACS Institute, which is supported by the National Center for Advancing Translational Sciences (NCATS), National Institutes of Health, through Grant Award Number UL1TR002489.

## Author contributions

D.M., S.J.W., and T.R.G. wrote the manuscript. D.M., S.J.W., J.K.O., B.B., T.D., A.H.C., A.M.D, R.V., and T.R.G. conducted the experiments and analyzed the data. K.W. and T.R.G. were responsible for the conception and oversight of the project. All authors discussed the results and reviewed the manuscript.

## Competing interests

The authors declare no competing interests.
