## [Peer Review File · Communications Biology]

Reviewers' comments:

Reviewer #1 (Remarks to the Author):

In this study by Malawsky et al., the authors build off previously published findings that demonstrate differences in medulloblastoma tumor progression depending on whether mice were engineered to express oncogenic SmoM2 in neural stem cells or committed progenitors. They show that while both types of medulloblastoma mouse models have similar transcriptomes as demonstrated by bulk analyses, the stem cell-driven SmoM2 tumors progressed faster and exhibited radioresistance, which is a novel finding. Progenitor-driven tumors contained fewer stem cells, were radio-sensitive and progressed slower. The authors also show differences between the 2 tumor types in terms of stromal populations; however, the biological and potential clinical relevance of this finding was not explored.

Overall, the authors provide some additional insight into differential expression profiles in stem cell vs. progenitor-driven tumors, with the radioresistance data being the most compelling. The fact that the stem cell-driven tumors are more resistant to radiation is interesting. In addition, the preliminary data on niche differences are promising. That being said, the major concern with the paper is the lack of novelty or original conclusions. While the study is technically sound and very well-written, a significant portion of the data recapitulate previously published findings which are frequently referenced by the authors (ie. Schüller et al., *Cancer Cell*, 2008; Zhang et al., *Cancer Cell* 2019, and the authors' recent publication Ocasio et al. in *Nature Communications*, 2019). This specifically pertains to the survival data, the stem cell signatures as well as the OLIG2+ cell profiles described in Figure 3. Specific comments are as follow:

1. Figure 1c-d: The RT data are interesting. Would it be possible to perform bulk or scRNA sequencing on treated M-Smo vs. G-Smo tumors to shed light on the mechanistic differences between them? Presumably, the higher stem cell proportion in these tumors would contribute to the RT resistance but additional insights into mechanism would help to further strengthen the manuscript.
2. Figure 3: What is the significance of the dynamic changes in the OLIG2+ cells in the M-Smo vs. the G-Smo treated tumors? Again, more mechanistic insight would strengthen the manuscript. For example, are there differences in proliferation of this cell population at the P15 timepoint?
3. The analyses of the tumor niche, particularly the myeloid cells, while interesting, is rather descriptive. What is the significance of the more diverse microglia/macrophage and fibroblast cells in the M-Smo tumors? The authors state that counting MHCII-expressing macrophages to distinguish between radiosensitive and radioresistant could be further explored in a clinical setting for prognostic purposes. If these differences can be probed in existing clinical samples by IHC or flow cytometry to provide validation of this hypothesis, the manuscript, in particular the novelty, would be significantly strengthened. For example, it would be very interesting to determine whether this pattern would hold between human samples representing the different SHH MB subtypes recently described by Cavalli et al., *Cancer Cell*, 2017.

Reviewer #2 (Remarks to the Author):

This work provides a careful transcriptomic analysis of mouse medulloblastomas initiated by constitutive activation of the Sonic Hedgehog pathway in Gfap-expressing cerebellar stem cells (G-Smo) versus cerebellar granule neuron progenitor (M-Smo). Despite having very similar pathological features, G-Smo and M-Smo tumors displayed significant differences in aggressiveness and response to radiotherapy. Bulk transcriptomic analysis revealed very few differences, but the characterization of

single cells expression profiles revealed in G-Smo tumors a higher frequency of OLIG2-expressing stem-like cells, as well as a lower frequency of macrophages. This is a very interesting paper, providing not only important new information on features that may help medulloblastoma prognostic assessment, but also underscoring the power of single cell analysis in obtaining crucial biological insight.

I have no general or technical remarks: the quality of data/data analysis conforms to high standards and the conclusions are well justified by data.

Authors may consider to explain a little better in the discussion why the bulk analysis does not show differences in the expression of proliferation markers (I suppose it is for their oscillatory nature) and of macrophagic markers (I would expect a difference, based on the increased representation revealed by the IF analysis). Moreover, they may consider to discuss the possibility that G-Smo and M-Smo tumors may functionally differ in their intrinsic DNA-repair capability, as an increased mutation burden could probably help bridging the increased radiosensitivity of M-Smo tumors with their increased macrophagic infiltrate.

However, I think that the manuscript is already acceptable in the current form.

Reviewer #3 (Remarks to the Author):

The approach by the authors to ask whether different outcomes in human SHH MBs may be determined by distinct cryptic events during development appears valid. Also, it is fine to compare therapy effects and tumor microenvironments in mouse models for MB that are driven by different promoters. Still, novel biological insights appear sparse and, above all, this paper clearly lacks validation in the human system, i.e. in human tumor samples or patients.

Other issues:

1. Use hGFAP-cre instead of Gfap-cre, if appropriate
2. Annotations and labelings of Figure 2 and 5 are too tiny. Please make any annotation in any Figure readable in a way that a regular print out of the paper is sufficient.
3. When comparing the expression of Olig2 in Figure 2f and on <https://gsmovmsmviewer.shinyapps.io/GvMviewer/>, things are not the same. On the website tool, there are a lot fewer Olig2 positive cells in the tumor. How come?
4. Please update ref 22, which has been published in the meantime
5. As for Figure 3, I don't really see the purpose of panels b-g. What's the message that the readers should get away with? What do the squares indicate (please indicate in the legends)? Is "a" a high power magnification of another image shown in this figure? Why not showing h at the end?
6. As for Figure 4, I cannot detect any green or brown circle as mentioned in the legend. Also, it is unclear to me, why so few tumor cells express YFP as shown in a. On the other hand, why should astrocytes and oligodendrocytes express YFP in Math1-cre::SmoM2 mice?
7. Please complete your descriptions by clearly writing down numbers of mice and experiments as well as statistical details wherever appropriate throughout the paper.

We thank the reviewers for their thorough and thoughtful reading of our manuscript. We have worked to address all of the reviewer comments, as described in the point-by-point response below. We feel that the manuscript is significantly strengthened by the new data added in response to the feedback and suggestions of the reviewers. We hope the reviewers agree.

We have included the reviewers' comments in italics. Our responses are in bold.

In this study by Malawsky et al., the authors build off previously published findings that demonstrate differences in medulloblastoma tumor progression depending on whether mice were engineered to express oncogenic SmoM2 in neural stem cells or committed progenitors. They show that while both types of medulloblastoma mouse models have similar transcriptomes as demonstrated by bulk analyses, the stem cell-driven SmoM2 tumors progressed faster and exhibited radioresistance, which is a novel finding.

We appreciate the reviewer recognizing the novelty of our data on differences in response to therapy, which are an important aspect of our study.

Progenitor-driven tumors contained fewer stem cells, were radio-sensitive and progressed slower. The authors also show differences between the 2 tumor types in terms of stromal populations; however, the biological and potential clinical relevance of this finding was not explored.

We have added new analyses to address the biological and potential clinical relevance, as we describe in the point-by-point response below. These changes include new data showing that :

- **fewer cells undergo apoptosis after radiation in G-Smo tumors,**
- **phosphorylated OLIG2 is increased in G-Smo tumors,**
- **more myeloid cells in G-Smo tumors express Igf1, and**
- **HLA-DR proteins, the human orthologs of mouse H2-Ea, show varied expression in patient-derived samples of different medulloblastoma subgroups and subtypes.**

Overall, the authors provide some additional insight into differential expression profiles in stem cell vs. progenitor-driven tumors, with the radioresistance data being the most compelling. The fact that the stem cell-driven tumors are more resistant to radiation is interesting. In addition, the preliminary data on niche differences are promising. That being said, the major concern with the paper is the lack of novelty or original conclusions. While the study is technically sound and very well-written, a significant portion of the data recapitulate previously published findings which are frequently referenced by the authors (i.e. Schüller et al., Cancer Cell, 2008; Zhang et al., Cancer Cell 2019, and the authors' recent publication Ocasio et al. in Nature Communications, 2019). This specifically pertains to the survival data, the stem cell signatures as well as the OLIG2+ cell profiles described in Figure 3.

Our findings that developmental timing of oncogenesis affects tumor stem cell regulation, tumor-TME I interactions and therapeutic outcomes are novel and are not contained within the papers cited above. In the revision, we have clarified the novelty of this finding, and added new data regarding mechanisms affected by the developmental timing of initial

oncogenic event. These changes have strengthened our conclusion that the developmental timing of oncogenesis is a key determinant of outcome.

Specific comments are as follow:

1. Figure 1c-d: The RT data are interesting. Would it be possible to perform bulk or scRNA sequencing on treated M-Smo vs. G-Smo tumors to shed light on the mechanistic differences between them? Presumably, the higher stem cell proportion in these tumors would contribute to the RT resistance but additional insights into mechanism would help to further strengthen the manuscript.

We have added new studies aimed to provide additional mechanistic information. Specifically, we analyzed treated tumors using bulk transcriptomic methods, we compared the cellular responses to radiation in *M-Smo* and *G-Smo* tumors, and we compared OLIG2 phosphorylation in untreated *M-Smo* and *G-Smo* tumors.

To address the reviewer's suggestion directly, we compared bulk transcriptomic data from irradiated *M-Smo* and *G-Smo* tumors. Our prior studies in *M-Smo* tumors show that a single fraction of xRT induces synchronous apoptosis throughout the tumor after a 3-hour latent period. We subjected *M-Smo* and *G-Smo* mice to a single dose of 10 Gy x-ray radiation, then harvested tumors 2 hours later, before the onset of apoptosis, purified RNA and performed microarray. We compared the 4 conditions, untreated *M-Smo*, untreated *G-Smo*, radiated *M-Smo* and radiated *G-Smo* using 2-way ANOVA. We detected statistically significant effects of genotype (genes differential in *G-Smo* vs *M-Smo*), and statistically significant effects of treatment (genes differential in radiated vs untreated), but we did not detect genotype-specific effects of treatment that were statistically significant. These data have been added to the Results section and the microarray data were uploaded to Geo. While these studies did not identify a molecular marker of treatment failure in *G-Smo* tumors, they confirm our underlying hypothesis that bulk transcriptomic studies can miss important underlying differences that require single-cell resolution to identify due to the effect of averaging across diverse cells.

We added important mechanistic detail in new studies of apoptosis after radiation therapy (xRT). These studies, described in the revised Fig. 1, show a significant difference in the apoptotic responses of *M-Smo* and *G-Smo* tumors to xRT. Our prior studies in *M-Smo* tumors show that apoptosis is required for radiation sensitivity. In a new experiment, we subjected *M-Smo* and *G-Smo* mice to 1 single 10Gy fraction of radiation and compared the fractions of apoptotic cells 4 hours after the radiation dose, using immunohistochemistry for cleaved caspase-3. Both genotypes showed apoptosis throughout the tumors, but quantitative analysis showed significantly fewer apoptotic cells in *G-Smo* tumors. The widespread induction of apoptosis in both genotypes is consistent with the similar average gene expression patterns demonstrated by bulk transcriptomic analysis, while the significantly larger populations of cells that do not undergo apoptosis in radiated *G-Smo* tumors suggest the presence of resistant populations that drive recurrence. These new data, provide a key link between the radiation outcome studies and the scRNA-seq finding of increased heterogeneity and stem cell populations in *G-Smo* tumors.

In an effort to provide mechanistic molecular information, we analyzed OLIG2 phosphorylation. Prior studies have shown that phosphorylated OLIG2 (pOLIG2) disrupts p53 function [1], and we, and others, have previously shown that p53 function is essential for apoptosis in response to radiation in SHH medulloblastoma [2]. Our pOLIG2 studies required a bulk approach by western blot, as all pOLIG2 antibodies that we tested failed in immunohistochemistry assays. Western blot effectively detected pOLIG2 and comparison between tumor genotypes showed increased pOLIG2 *G-Smo* tumors versus *M-Smo* tumors. We have added these data, which highlight a potential mechanism for radio-resistance in *G-Smo* tumors, to the manuscript and to the revised Fig.3.

2. *Figure 3: What is the significance of the dynamic changes in the OLIG2+ cells in the M-Smo vs. the G-Smo treated tumors?*

We revised the text to clarify the significance of the dynamic changes by stating “The difference in the developmental timing of oncogenesis in *G-Smo* and *M-Smo* tumors thus continued to affect stem cell dynamics weeks after the oncogenic events.”

Again, more mechanistic insight would strengthen the manuscript. For example, are there differences in proliferation of this cell population at the P15 timepoint?

We appreciate this question. To provide an answer, we analyzed proliferation rate in the OLIG2+ population of *G-Smo* and *M-Smo* tumors by performing double-label immunohistochemistry for OLIG2 and proliferation marker phosphorylated RB (pRB). We found no significant difference in the fractions of OLIG2+ cells that were pRB+ in *G-Smo* vs *M-Smo* tumors. We included the finding that the proliferation rates in the stem cell populations are similar in the revised manuscript as Supplementary Fig. 2. Based on these data, we conclude that differences in stem cell populations between *G-Smo* and *M-Smo* tumors are due to differences in stem cell maintenance, with stem cells more likely to lose their stem cell character over time in *M-Smo* tumors.

3. *The analyses of the tumor niche, particularly the myeloid cells, while interesting, is rather descriptive. What is the significance of the more diverse microglia/macrophage and fibroblast cells in the M-Smo tumors?*

We added a new analysis that demonstrates a significant difference in the *Igf1*+ myeloid cells of *G-Smo* and *M-Smo* tumors. We undertook this analysis in light of a recent paper showing myeloid cells in SHH medulloblastoma secrete IGF1, and that this paracrine signaling is required for medulloblastoma progression [3] and we have added this reference to the revised manuscript. In our new analysis, we confirmed that *Igf1* mRNA was expressed specifically by myeloid cells in both *G-Smo* and *M-Smo* tumors, as in the previously reported model. Importantly, we found that the different subsets of myeloid cells in *G-Smo* and *M-Smo* tumors showed differential *Igf1* expression. The cells of the myeloid cluster ^M2, comprising *Ccr2*+ macrophages, expressed significantly lower *Igf1*, and these cells were specifically enriched in *M-Smo* tumors. In contrast, myeloid cells in *G-Smo* tumors were predominantly *Igf1*-expressing subtypes. Direct comparison confirmed that *G-Smo* tumors contained significantly more *Igf1*+ myeloid cells, compared to *M-Smo* tumors. We have added our new analysis to the Results section of the revision and to the revised Figure 6.

As we discuss in the revised Discussion section, different SHH medulloblastoma studies have identified tumor suppressive and tumor supportive functions of myeloid cells [4, 5]. Our new finding that *Igf1*⁺ and *Igf1*⁻ subsets vary in SHH-driven medulloblastomas with different progression patterns provides a mechanistic framework for understanding how differences in myeloid populations can alter outcomes.

The authors state that counting MHCII-expressing macrophages to distinguish between radiosensitive and radioresistant could be further explored in a clinical setting for prognostic purposes. If these differences can be probed in existing clinical samples by IHC or flow cytometry to provide validation of this hypothesis, the manuscript, in particular the novelty, would be significantly strengthened. For example, it would be very interesting to determine whether this pattern would hold between human samples representing the different SHH MB subtypes recently described by Cavalli et al., Cancer Cell, 2017.

Validating biomarkers in clinical samples is a resource intensive task, as medulloblastoma samples stratified Cavalli subtypes are a scarce resource and establishing immunohistochemistry assays for many antigens can be unpredictable. Recognizing these challenges, we conducted a pilot study as suggested by the reviewer. We recruited additional collaborators, Dr. Rajeev Vibhakar and Dr. Andrew Donson (both at University of Colorado). We examined HLA-DR proteins, the human orthologs of the mouse MHCII protein H2-EA, in a set of medulloblastoma samples resected from patients and subjected to DNA methylomic analysis by Drs. Vibhakar and Donson to determine sub-type as in Cavalli et al., Cancer Cell, 2017. Our sample included 30 medulloblastomas, of which 10 were SHH subgroup and 9 could be further classified by SHH subtype. These 9 SHH medulloblastomas included 2 SHH alpha subtype, 4 SHH beta subtype and 3 of SHH delta subtype. We successfully stained paraffin sections of each tumor for HLA-DR and then scored HLA-DR staining on a scale of 0-3 in a blinded analysis. Samples of each score are shown in the new Figure 7.

We found significantly increased HLA-DR in the SHH subgroup tumors compared to the other subgroups, demonstrated by higher HLA-DR scores. Between SHH subtypes, we noted similar, relatively higher scores in the SHH beta and delta subtypes, and markedly lower scores in the SHH alpha subtype. These new data are presented in the revised Fig. 7. While a statistical analysis of SHH subtypes will require more samples and resources beyond the scope of this project, we point out that “SHH beta and SHH delta are the infant-predominant subtypes [6], and the trend toward higher HLA-DR expression in these subtypes suggests that a developmental process affects myeloid subgroups in human tumors, as in our mouse models”.

To increase the statistical power using available, published data, we added a new analysis of *HLA-DRA* mRNA expression in the published dataset from Cavalli et al [6]. These data also show increased expression in SHH subgroup tumors and within the SHH subgroup show that SHH alpha tumors show lower average expression compared the other SHH subtypes. Moreover, analysis of SHH subgroup tumors by age, rather than subtype show significantly higher *HLA-DRA* in infant-onset SHH medulloblastomas. While the differences in average expression are statistically significant, we note large individual variation in each subtype, suggesting heterogeneity within each subtype. We suggest in the

Discussion that additional large studies of clinical samples using cell-resolved methods such as scRNA-seq or IHC may allow the definition of more homogeneous subsets.

Reviewer #2 (Remarks to the Author):

This work provides a careful transcriptomic analysis of mouse medulloblastomas initiated by constitutive activation of the Sonic Hedgehog pathway in Gfap-expressing cerebellar stem cells (G-Smo) versus cerebellar granule neuron progenitor (M-Smo). Despite having very similar pathological features, G-Smo and M-Smo tumors displayed significant differences in aggressiveness and response to radiotherapy. Bulk transcriptomic analysis revealed very few differences, but the characterization of single cells expression profiles revealed in G-Smo tumors a higher frequency of OLIG2-expressing stem-like cells, as well as a lower frequency of macrophages. This is a very interesting paper, providing not only important new information on features that may help medulloblastoma prognostic assessment, but also underscoring the power of single cell analysis in obtaining crucial biological insight.

I have no general or technical remarks: the quality of data/data analysis conforms to high standards and the conclusions are well justified by data.

We appreciate that the reviewer considered the conclusions were well justified.

Authors may consider to explain a little better in the discussion why the bulk analysis does not show differences in the expression of proliferation markers (I suppose it is for their oscillatory nature) and of macrophagic markers (I would expect a difference, based on the increased representation revealed by the IF analysis).

We added a paragraph to the Discussion section that considers factors that may obscure important signals in bulk transcriptomic studies. We attribute the lack of significant differences in proliferation markers in the bulk studies to the high levels of proliferative cells in all groups, which may prevent proliferation marker mRNAs from showing statistically detectable variation. We attribute the lack of different signals from myeloid subsets as related to their relatively small fractions in the tumors, and we attribute the lack of differences in *Olig2* signals to the common expression of *Olig2* by both stem cells and oligodendrocytes.

Moreover, they may consider to discuss the possibility that G-Smo and M-Smo tumors may functionally differ in their intrinsic DNA-repair capability, as an increased mutation burden could probably help bridging the increased radiosensitivity of M-Smo tumors with their increased macrophagic infiltrate.

We appreciate the suggestion that a difference in DNA repair capability may contribute to the difference in radiosensitivity, which is consistent with our new data showing increased pOLIG2 expression in G-Smo tumors. We discuss the possibility that inhibition of p53-dependent apoptosis by pOLIG2 may allow increased time for stem cells to repair DNA after xRT. We cite a prior paper showing that *Bax*-deleted medulloblastomas which are apoptosis incompetent, repair DNA breaks after xRT.

However, I think that the manuscript is already acceptable in the current form.

Reviewer #3 (Remarks to the Author):

The approach by the authors to ask whether different outcomes in human SHH MBs may be determined by distinct cryptic events during development appears valid. Also, it is fine to compare therapy effects and tumor microenvironments in mouse models for MB that are driven by different promoters.

We used Cre driven by different promoters to initiate conditional *SmoM2* expression at different points in development. In each model, *SmoM2* is driven by the same endogenous promoter of the *Rosa26* locus. *SmoM2* is expressed in overlapping sets of cells in *G-Smo* and *M-Smo* mice, starting either early (*G-Smo*) mice or later (*M-Smo*) in the lineage that extends from multipotent stem cells to committed neural progenitors. We revised the text to clarify this distinction.

Still, novel biological insights appear sparse

The revised text includes new mechanistic data that connect our observations of radioresistance and increased OLIG2+ stem cells in *G-Smo* tumors. The revised manuscript better emphasizes our primary biological insight, which is that cellular heterogeneity and stem cell maintenance can be determined by the timing of initial oncogenic event and can, in turn, act as determinants of clinical outcomes. The new Figure 1 includes data showing that a large fraction of *G-Smo* tumors escape apoptosis after radiation, and the revised Figure 3 shows that pOLIG2, which is known to disable p53, is increased in *G-Smo* tumors. The revised Figure 6 shows that *Igf1*, which provides essential paracrine support in medulloblastoma, is increased in *G-Smo* tumors and correlates with increased mTORC1 activation (demonstrated by increased p4EBP1). These new data support a model in which OLIG2+ stem cells, which in *G-Smo* tumors are more numerous and better supported by *Igf1*+ myeloid cells, evade p53-dependent apoptosis and drive recurrence. This model provides a mechanistic framework connecting radioresistance to developmental differences.

and, above all, this paper clearly lacks validation in the human system, i.e. in human tumor samples or patients.

In response to this comment, as well as comments of Reviewer 1, we have added new data derived from a collaboration with additional investigators, now added to the authorship, in which we study HLA-DR protein expression in patient-derived medulloblastoma samples. We also collaborated on a new analysis of *HLA-DRA* mRNA expression in published bulk transcriptomic data from medulloblastoma patients. These new data validate in the human system specific observations made in our genetic models.

Other issues:

1. Use hGFAP-cre instead of Gfap-cre, if appropriate

We have changed the text to read hGFAP-Cre, as suggested.

2. Annotations and labeling of Figure 2 and 5 are too tiny. Please make any annotation in any figure readable in a way that a regular print out of the paper is sufficient.

We appreciate the suggestion revise the figure annotations and labels. We have addressed this issue in each figure, and the revision has greatly increased the readability.

3. When comparing the expression of Olig2 in Figure 2f and on <https://gsmovmsmoviewer.shinyapps.io/GvMviewer/>, things are not the same. On the website tool, there are a lot fewer Olig2 positive cells in the tumor. How come?

The viewer allows the user to change the threshold signal defined as positive. This adjustment is important, as different genes show different ranges of mRNA copies detected in positive cells, the number of mRNAs that must be expressed for biological effects varies depending on the gene and most often this number is not known. By adjusting the threshold, the reviewer can reproduce the image of Olig2 in Figure 2f.

4. Please update ref 22, which has been published in the meantime

We have made the update.

5. As for Figure 3, I don't really see the purpose of panels b-g. What's the message that the readers should get away with?

We have revised the presentation of the data in Figure 3. The revised text clarifies that panels that were labelled b-g are representative images provided to give the readers a sense of the spatial distribution of OLIG2+/SOX10- cells throughout the tumors. In the revision, these panels as a group are labeled 3b and the results are quantified in the adjacent panel 3c. These visual data are needed because the spatial information is removed from the quantitative analysis of panel 3c.

What do the squares indicate (please indicate in the legends)?

The squares should not have been included in the figure and they have been removed.

Is "a" a high power magnification of another image shown in this figure?

We revised the Figure Legend to state clearly that 3a is a high-power image from a P15 *G-Smo* tumor, provided to show the different labelling patterns of OLIG2 and SOX10 antibodies.

Why not show h at the end?

This suggestion is taken into account in the new Figure 3, in which the quantification is now 3c.

6. As for Figure 4, I cannot detect any green or brown circle as mentioned in the legend.

We have corrected the figure.

Also, it is unclear to me, why so few tumor cells express YFP as shown in a.

Different genes produce different ranges of mRNA abundance, and *SmoM2-Yfp* is a low abundance gene. For genes that are not highly abundant in positive cells, the risk of false negatives in scRNA-seq is higher. This issue is a recognized limitation of scRNA-seq, that

we discussed in our prior work that described the method of *SmoM2-Yfp* lineage tracing. We cite this work in the description of the data.

*On the other hand, why should astrocytes and oligodendrocytes express YFP in *Math1-cre::SmoM2* mice?*

We [7] and others[3] have shown that tumor cells in SHH medulloblastoma trans-differentiate to generate glial cells. This trans-differentiation is tumor-specific and does not occur in normal SHH-driven proliferation during cerebellar development. The revised text includes these citations to document this trans-differentiation process.

7. Please complete your descriptions by clearly writing down numbers of mice and experiments as well as statistical details wherever appropriate throughout the paper.

We have revised the text to state the the numbers of replicates and the text and figure legends to identify clearly the statistical tests used. The number of replicates is also shown in each graph, as each point represents an individual replicate.

Works Cited in this Letter

1. Mehta, S., et al., *The central nervous system-restricted transcription factor Olig2 opposes p53 responses to genotoxic damage in neural progenitors and malignant glioma*. *Cancer Cell*, 2011. **19**(3): p. 359-71.
2. Crowther, A.J., et al., *Radiation Sensitivity in a Preclinical Mouse Model of Medulloblastoma Relies on the Function of the Intrinsic Apoptotic Pathway*. *Cancer Res*, 2016. **76**(11): p. 3211-23.
3. Yao, M., et al., *Astrocytic trans-Differentiation Completes a Multicellular Paracrine Feedback Loop Required for Medulloblastoma Tumor Growth*. *Cell*, 2020. **180**(3): p. 502-520 e19.
4. Maximov, V., et al., *Tumour-associated macrophages exhibit anti-tumoural properties in Sonic Hedgehog medulloblastoma*. *Nat Commun*, 2019. **10**(1): p. 2410.
5. Lee, C., et al., *M1 macrophage recruitment correlates with worse outcome in SHH Medulloblastomas*. *BMC Cancer*, 2018. **18**(1): p. 535.
6. Cavalli, F.M.G., et al., *Intertumoral Heterogeneity within Medulloblastoma Subgroups*. *Cancer Cell*, 2017. **31**(6): p. 737-754 e6.
7. Ocasio, J., et al., *scRNA-seq in medulloblastoma shows cellular heterogeneity and lineage expansion support resistance to SHH inhibitor therapy*. *Nat Commun*, 2019. **10**(1): p. 5829.

REVIEWERS' COMMENTS:

Reviewer #1 (Remarks to the Author):

I appreciate the concerted effort made by the authors to address my questions. I am satisfied with the changes and have no further major comments. However, I would suggest shortening the paper, particularly the results section, as it is quite long and would benefit from being more concise.

Reviewer #3 (Remarks to the Author):

No further comments. Questions from this reviewer have been answered appropriately.